# ARIH1 activates STING-mediated T-cell activation and sensitizes tumors to immune checkpoint blockade

Xiaolan Liu[1,14], Xufeng Cen[2,14], Ronghai Wu[3,14], Ziyan Chen[4,14], Yanqi Xie[5,14], Fengqi Wang[1], Bing Shan[6], Linghui Zeng[7], Jichun Zhou [8], Bojian Xie[9], Yangjun Cai[9], Jinyan Huang [10], Yingjiqiong Liang[10], Youqian Wu[11], Chao Zhang[12], Dongrui Wang[2] & Hongguang Xia [1,2,13] ✉

Despite advances in cancer treatment, immune checkpoint blockade (ICB) only achieves complete response in some patients, illustrating the need to identify resistance mechanisms. Using an ICB-insensitive tumor model, here we discover cisplatin enhances the anti-tumor effect of PD-L1 blockade and upregulates the expression of Ariadne RBR E3 ubiquitin-protein ligase 1 (ARIH1) in tumors. *Arih1* overexpression promotes cytotoxic T cell infiltration, inhibits tumor growth, and potentiates PD-L1 blockade. ARIH1 mediates ubiquitination and degradation of DNA-PKcs to trigger activation of the STING pathway, which is blocked by the phospho-mimetic mutant T68E/S213D of cGAS protein. Using a high-throughput drug screen, we further identify that ACY738, less cytotoxic than cisplatin, effectively upregulates ARIH1 and activates STING signaling, sensitizing tumors to PD-L1 blockade. Our findings delineate a mechanism that tumors mediate ICB resistance through the loss of ARIH1 and ARIH1-DNA-PKcs-STING signaling and indicate that activating ARIH1 is an effective strategy to improve the efficacy of cancer immunotherapy.

Tumors escape the surveillance of the immune system by a variety of mechanisms during their occurrence and development[1,2]. Immune checkpoint blockade (ICB) is the main pathway involved in tumor escape[1]. The use of immune checkpoint inhibitors, such as programmed cell death 1 (PD-1), programmed cell death ligand-1 (PD-L1),

and cytotoxic T-lymphocyte associated protein 4 (CTLA-4), to treat tumors has shown great potential. The fraction of patients that benefit from ICB is low, with response rates of approximately 20–40%[3], highlighting that strategies are needed to enhance the anti-tumor efficacy of ICB. Positive PD-L1 expression has been reported as a

[1]Department of Biochemistry & Research Center of Clinical Pharmacy of The First Affiliated Hospital, Zhejiang University School of Medicine, Hangzhou 310058, China. [2]Liangzhu Laboratory, Zhejiang University Medical Center, 1369 West Wenyi Road, Hangzhou 311121, China. [3]Hangzhou PhecdaMed Co.Ltd, 2626 Yuhangtang Road, Hangzhou 311121, China. [4]Department of Urology, the First Affiliated Hospital, Zhejiang University School of Medicine, Hangzhou 310003, China. [5]Department of Urology, the Second Affiliated Hospital, Zhejiang University School of Medicine, Hangzhou 310009, China. [6]Interdisciplinary Research Center on Biology and Chemistry, Shanghai Institute of Organic Chemistry, Chinese Academy of Sciences, Shanghai 201203, China. [7]Key Laboratory of Novel Targets and Drug Study for Neural Repair of Zhejiang Province, School of Medicine, Hangzhou City University, 50# Huzhou Rd., Hangzhou, Zhejiang, China. [8]Department of Surgical Oncology, Affiliated Sir Run Run Shaw Hospital, Zhejiang University School of Medicine, Hangzhou, Zhejiang 310000, China. [9]Affiliated Taizhou Hospital, Wenzhou Medical University, Taizhou 318000, China. [10]Biomedical big data center, the First Affiliated Hospital, Zhejiang University School of Medicine, 79 Qingchun Road, Hangzhou 310003, China. [11]International Institutes of Medicine, The Fourth Affiliated Hospital, Zhejiang University School of Medicine, Yiwu, Zhejiang, China. [12]Department of Anatomy and Department of Respiratory and Critical Care Medicine, the Second Affiliated Hospital, Zhejiang University School of Medicine, Hangzhou, Zhejiang 310009, China. [13]Cancer Center, Zhejiang University, Hangzhou 310058, China. [14]These authors contributed equally: Xiaolan Liu, Xufeng Cen, Ronghai Wu, Ziyan Chen, Yanqi Xie. ✉e-mail: hongguangxia@zju.edu.cn

potential predictor of response to immunotherapy[4–6]. Although some triple-negative breast cancer (TNBC) patients have high PD-L1 expression, they are accompanied by high immunosuppression in clinical trials with PD-1 or PD-L1 blockade, as reflected by low levels of T-cell infiltration and low clinical response rates[7–9]. Many human cancers fail to respond to ICB therapy, and these resistant tumors frequently have poor infiltration by CD8+ T cells (cold tumor properties)[10]. Recent studies have demonstrated that the low-dose chemotherapeutic agent cisplatin enhances anti-PD-1/PD-L1 immunotherapy efficacy to inhibit resistant tumor growth[11–13], but the mechanisms are not understood.

ARIH1 (also named HHARI), is one member of the Ring-Between-Ring (RBR) subfamily of E3 ubiquitin ligases and plays an essential role in regulating protein translation and RNA processing by promoting 4EHP ubiquitination in response to DNA damage[14]. Moreover, ARIH1 was shown to polyubiquitinate damaged mitochondria, leading to their clearance by autophagy[15]. Suppressed ARIH1 reduced PD-L1 degradation in lung adenocarcinoma biopsies, contributing to cancer escape from anti-tumor immunity[16]. However, it is unclear whether ARIH1 regulates tumor immunity by participating in other pathways rather than PD-L1 degradation.

In this study, we uncover that ARIH1 regulates DNA-PKcs ubiquitination and degradation, thereby activating the intrinsic STING pathway in tumor cells to promote anti-tumor immunity. Increasing ARIH1 levels by genetic overexpression or pharmacological agents reactivate CD8+ T cells to sensitize tumors to PD-L1 blockade therapy. Analysis of animal and clinical data shows a positive correlation between ARIH1 expression and checkpoint blockade response in multiple tumors, suggesting that ARIH1 is a promising target for tumor immunotherapy.

## RESULTS

### Cisplatin increases ARIH1 protein levels and enhances PD-L1 blockade efficacy

Poor infiltration of CD8+ T cells correlated with lower overall survival (OS) of patients in a number of human cancer types (Supplementary Fig. 1a), and contributed to the resistance to anti-PD-L1 therapy[17–19]. To model the "cold tumor" properties, we used 4T1-derived murine TNBC models which has been demonstrated insensitive to immunotherapy[20]. To examine whether cisplatin could enhance the efficacy of PD-L1 mAb treatment, we treated the 4T1 TNBC model with cisplatin and/or anti-PD-L1 (Supplementary Fig. 1b). As expected, anti-PD-L1 alone did not decrease tumor growth compared with IgG control. However, a significant decrease in tumor growth was observed by combining anti-PD-L1 with cisplatin (Fig. 1a, b, Supplementary Fig. 1c). The survival of the combinational treatment group was also greatly increased (Fig. 1c). We then established a lung metastasis model, inoculating 4T1 cells intravenously which preferentially led to tumor lesions in the lungs. In this model, tumor-bearing mice treated with anti-PD-L1 and cisplatin resulted in fewer tumor nodules in lungs compared to anti-PD-L1 or IgG control alone (Supplementary Fig. 1d, e). The levels of CD8+ and GzmB+CD8+ T cells were significantly increased in tumors from the combination treatment group compared with monotherapy groups, as shown by both flow cytometry (Supplementary Fig. 1f, g, Supplementary Fig. 2) and immunohistochemistry (IHC) staining (Fig. 1d, e). These data collectively suggest that cisplatin enhances the efficacy of PD-L1 blockade and induces CD8+ T cell activation on resistant tumors.

In accordance with the previous report[14], cisplatin treatment resulted in a significant increase in intracellular and intratumoral ARIH1 levels (Fig. 1f, g, Supplementary Fig. 3), indicating that ARIH1 may participate in enhancing anti-tumor T-cell immunity in the combination group. To test whether ARIH1 is required for the anti-tumor effects of cisplatin-enhanced PD-L1 blockade therapy, we tested the therapeutic efficacy against a subcutaneously tumor model established with *Arih1*-knockdown 4T1 cells. We found that with

*Arih1*-knockdown, tumors progressed even with combined cisplatin and PD-L1 blockade therapy (Fig. 1h–j, Supplementary Fig. 4a, b). Consistently, tumors with *Arih1* knockdown showed no increase in CD8+ T cell or GzmB+CD8+ T cell infiltration after combination therapy (Supplementary Fig. 4c, d). Therefore, the superior antitumor potency of combining cisplatin with anti-PD-L1 is dependent on ARIH1 expression on tumor cells.

### ARIH1 promotes anti-tumor immunity and improves ICB efficacy

Having demonstrated that ARIH1 knockdown disrupted the antitumor effect of cisplatin-anti-PD-L1 combinational therapy, we next sought to evaluate whether increasing ARIH1 levels could promote the efficacy of PD-L1 blockade therapy. We established a 4T1 cell line with stable *Arih1-WT*-OE (*Arih1* overexpression). The tumor-bearing mice were randomly divided into 4 groups and treated with IgG control or PD-L1 mAbs for 2 weeks. We discovered that tumor growth in the *Arih1-WT*-OE group was significantly decreased compared with wild-type control, and treating with anti-PD-L1 led to a further decrease in tumor burden and achieved complete regression (Fig. 2a–c). However, *Arih1-WT*-OE had no effect on tumor growth in immunodeficient nude mice (Supplementary Fig. 5b, c). In addition, in vitro proliferation of 4T1, E0771, and B16-F10 cells was not affected by *Arih1* overexpression (Supplementary Fig. 5a). The survival of *Arih1-WT*-OE tumor-bearing mice treated with anti-PD-L1 was also significantly prolonged compared to the wild-type group treated with IgG or anti-PD-L1 (Fig. 2k). Meanwhile, *Arih1-WT*-OE tumors also showed increased numbers of infiltrating CD8+ T cells and GzmB+CD8+ T cells, which was further induced by anti-PD-L1 (Fig. 2d–g). These data collectively show that *Arih1-WT*-OE in tumor cells can induce CD8+ T cell activation and improve the efficacy of PD-L1 blockade.

To further investigate whether the enhanced T cell activation was dependent on the E3 ligase activity of ARIH1, we introduced a C355S mutation of ARIH1, which is analogous to the human E3 ligase-dead protein (ARIH1-C357S). Treating *Arih1-C355S*-OE tumors with anti-PD-L1 showed no reduction in tumor growth or significant survival benefit (Fig. 2h–k). Consistently, *Arih1-C355S*-OE tumors treated with anti-PD-L1 did not cause the accumulation of tumor-infiltrating CD8+ and GzmB+CD8+ T cells (Fig. 2l, m).

Similar phenotypes were observed in two other models, including E0771 and B16-F10 (Supplementary Fig. 6a–j). Collectively, these data suggest that ARIH1-WT-OE enhances PD-L1 blockade-induced anti-tumor immunity dependent on its enzymatic activity.

We next explored whether ARIH1 expression positively correlated with tumor response to checkpoint blockade therapy in mouse models. We compared the protein levels of ARIH1 in six murine cancer cell lines and found that B16-F10 cells had relatively high levels of ARIH1 (ARIH1-High), while 4T1 had relatively low levels of ARIH1 (ARIH1-Low) (Supplementary Fig. 7a). Subsequently, we tested the response of two types of tumor cells to PD-L1 blockade in syngeneic mouse models. Compared to 4T1 tumors (ARIH1-Low), treatment of B16-F10 tumors (ARIH1-High) with PD-L1 blockade greatly reduced tumor growth and resulted in a significant survival benefit, consistent with the sensitivity to PD-L1 checkpoint blockade observed clinically in patients with advanced melanoma[21, 22] (Supplementary Fig. 7b, c). In addition, we performed single-cell sorting of the same type of tumor cells (4T1 and E0771) and selected monoclonal cells with relatively high and low ARIH1 protein levels, respectively, to test their response to PD-L1 blockade. We found that, in the same type of tumor cells, mice with tumors with relatively high ARIH1 expression (ARIH1-High) were more sensitive to PD-L1 treatment, with substantially reduced tumor growth and higher tumor inhibition rate compared to the group with relatively low ARIH1 expression (ARIH1-Low) (Supplementary Fig. 7d–k).

We also examined the relevance of ARIH1 gene expression in patients enrolled in clinical ICB studies. We found that ICB responders

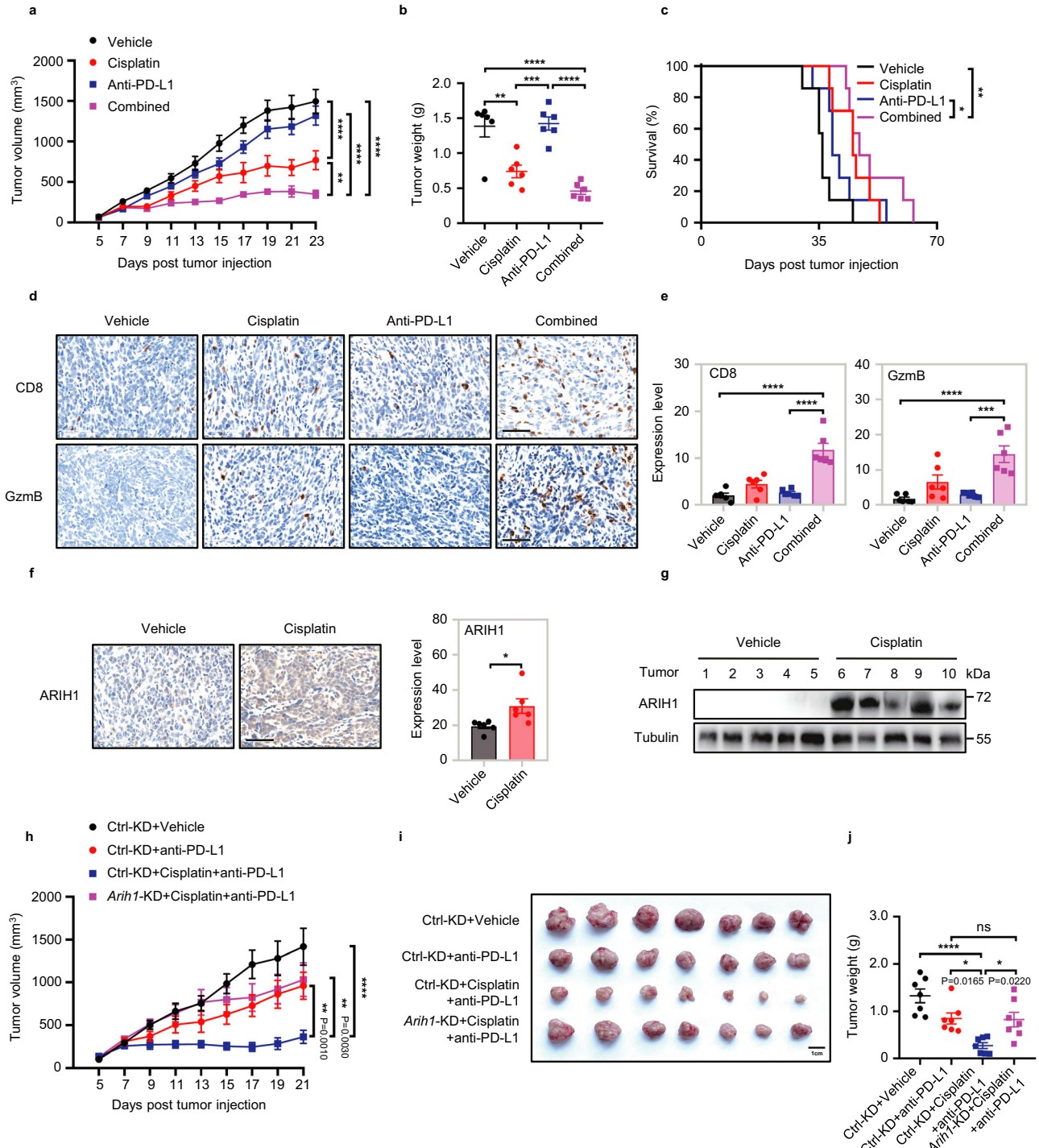

**Fig. 1 | ARIH1 accumulates after treatment of 4T1 tumor models with a combination of cisplatin and anti-PD-L1 antibody. a, b** Tumor growth curves and tumor weights upon subcutaneous injection of $5 \times 10^5$ 4T1 cells into female BALB/c mice (6–8 week old) treated with vehicle, cisplatin alone, anti-PD-L1 alone and cisplatin+anti-PD-L1. $n = 6$ mice/group. Data represent means ± SEM. **a** $^{**}P < 0.01$ ($P = 0.0052$), $^{****}P < 0.0001$. **b** $^{**}P < 0.01$ ($P = 0.0014$), $^{***}P < 0.001$ ($P = 0.0007$), $^{****}P < 0.0001$. **c** The survival curves of tumor bearing mice with indicated treatments. $n = 7$ mice/group. Log-rank test, $^{*}P < 0.05$ ($P = 0.0351$), $^{**}P < 0.01$ ($P = 0.0023$). **d, e**. Representative images of tumor CD8 and GzmB IHC staining of the mice as in (**a**). The percent of each expression pattern was quantified (**e**). Scale bar, 60 μm. $n = 6$ mice/group. Data represent means ± SEM, $^{***}P < 0.001$ ($P = 0.0002$),

$^{****}P < 0.0001$. **f** Representative ARIH1 IHC staining for tumors of the mice as in (**a**). The percent of expression pattern was quantified. Scale bar, 60 μm. $n = 6$ mice/group. Data represent means ± SEM, $^{*}P < 0.05$ ($P = 0.0233$). **g** Immunoblots analysis of the ARIH1 protein levels in the indicated tumor lysates of mice as in (**a**). **h–j** Tumor growth, final tumor image and tumor weights of Ctrl-KD and *Arih1*-KD 4T1 cells in female BALB/c mice ($n = 7$ per group, 6–8 week old) with indicated treatments. Data represent means ± SEM. **h** $^{**}P < 0.01$, $^{****}P < 0.0001$. **j** $^{*}P < 0.05$, $^{****}P < 0.0001$, ns, not significant. Data shown in **a** and **h** are representative of three independent experiments. For **a, h** data, Two-way ANOVA test. For **b, e, j** data, One-way ANOVA test. For **f** data, Two-tailed *t*-test. Source data are provided as a Source Data file.

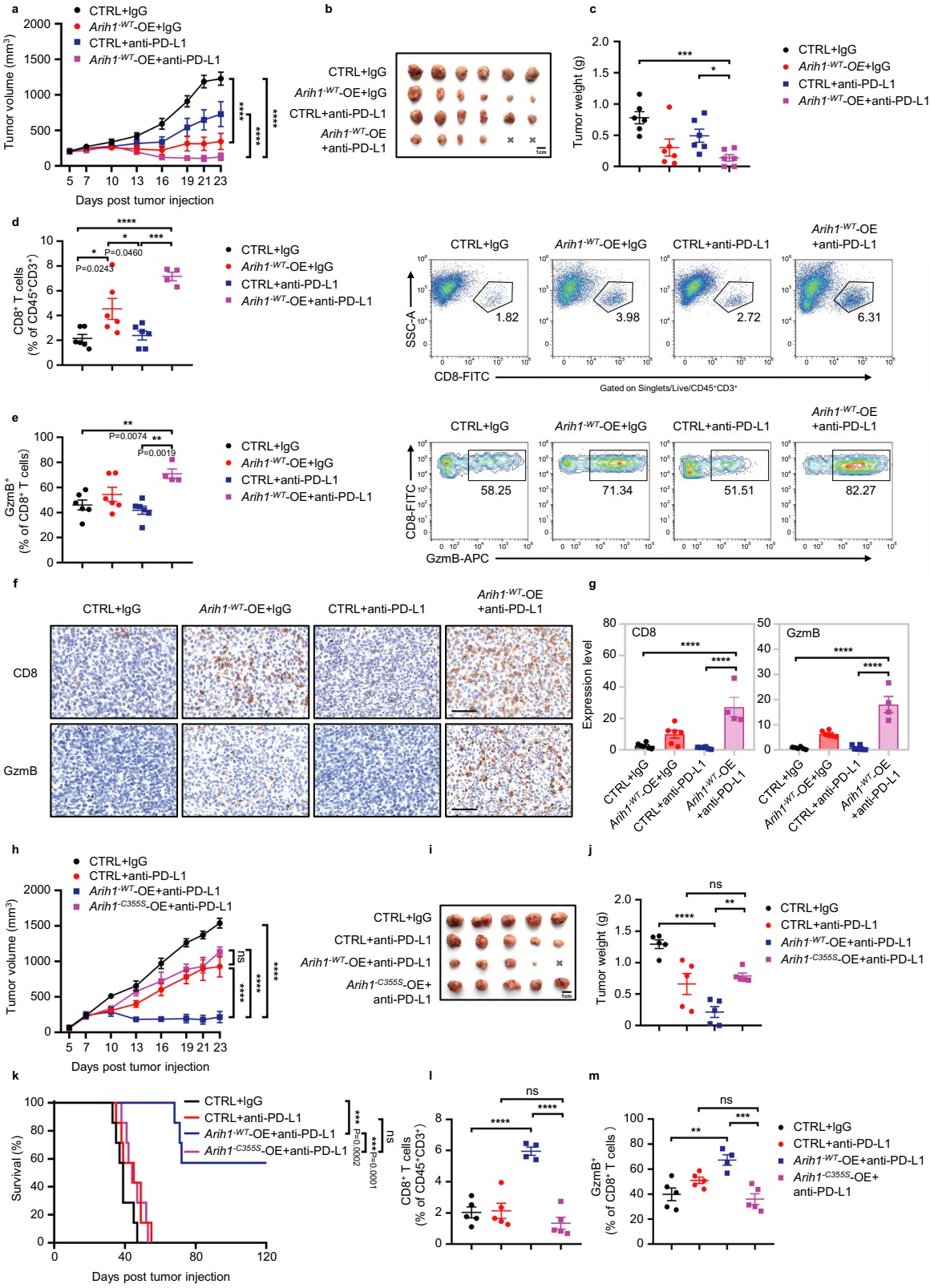

had higher expression levels of ARIH1 compared to non-responders at on-treatment time points (Supplementary Fig. 8a). Moreover, high ARIH1 mRNA levels (ARIH1[+]) were also associated with improved overall survival (OS) and progression-free survival (PFS) in multiple cancer types treated with ICB (Supplementary Fig. 8b, c), suggesting that patients with ARIH1[+] tumors may have a better response to ICB. These results suggest a positive correlation between ARIH1 expression

and tumor response to checkpoint blockade therapy in both mouse models and patients.

## The STING pathway is required for ARIH1 enhancing PD-L1 blockade-induced anti-tumor immunity

To explore the molecular mechanisms how ARIH1 activates CD8[+] T cells and enhances PD-L1 blockade therapy, we performed RNA

**Fig. 2 | ARIH1 enhances anti-PD-L1 antibody-induced anti-tumor immunity in 4T1 tumor models. a** Tumor growth curves in female BALB/c mice (6–8 week old) with control (CTRL) and *Arih1*-overexpressing (*Arih1-WT*-OE) tumors treated with PD-L1 or isotype mAbs intraperitoneally (i.p.) starting on day 7 and then every three days after subcutaneous inoculation of $5 \times 10^5$ 4T1 cells. *n* = 6 mice/group. Data represent means ± SEM, ****$P$ < 0.0001. **b, c** Representative image of tumors and tumor weights of tumor bearing mice at Day25 with the indicated treatments. *n* = 6 mice/group. Data represent means ± SEM, *$P$ < 0.05 ($P$ = 0.0134), ***$P$ < 0.001 ($P$ = 0.0002). **d, e** Representative figures and quantification of tumor infiltrating CD8$^+$ T cells and CD8$^+$GzmB$^+$ T cells of the mice as in **a**. *n* = 6, 6, 6, 4 mice/group. Data represent means ± SEM. **d** *$P$ < 0.05, ***$P$ < 0.001 ($P$ = 0.0001), ****$P$ < 0.0001. **e** **$P$ < 0.01. **f, g** Representative images of tumor CD8 and GzmB IHC staining of mice as in **a**. The percent of each expression pattern was quantified (**g**). Scale bar, 60 µm.

*n* = 6, 6, 6, 4 mice/group. Data represent means ± SEM, ****$P$ < 0.0001. **h–j** Representative tumor growth, image of tumors, and tumor weights in female BALB/c mice (6–8 week old) bearing CTRL, *Arih1-WT*-OE, and *Arih1-C3SSS*-OE 4T1 tumors with the indicated treatments. *n* = 5 mice/group. Data represent means ± SEM. **h** ****$P$ < 0.0001, ns, not significant. **j** **$P$ < 0.01 ($P$ = 0.0059), ****$P$ < 0.0001, ns, not significant. **k.** The survival curves for mice (*n* = 7 per group) with the indicated treatments. Log-rank test, ***$P$ < 0.001, ns, not significant. **l, m** Quantification of FACS data for tumor infiltrating CD8$^+$ T cells and CD8$^+$GzmB$^+$ T cells of mice as in **h**. *n* = 5, 5, 4, 5 mice/group. Data represent means ± SEM, **$P$ < 0.01 ($P$ = 0.0020), ***$P$ < 0.001 ($P$ = 0.0006), ****$P$ < 0.0001, ns, not significant. Data shown in **a** and **h** are representative of three independent experiments. For **a, h** data, Two-way ANOVA test. For **d, e, g, j, l, m** data, One-way ANOVA test. For **c** data, Two-tailed *t*-test. Source data are provided as a Source Data file.

sequencing (RNA-seq) to determine whether *Arih1-WT*-OE affects the gene expression in 4T1 cells. Gene set enrichment analysis (GSEA) revealed that the gene sets related to STING pathway were upregulated in *Arih1-WT*-OE cells compared to *Arih1-C3SSS*-OE cells, indicating that *Arih1-WT*-OE in tumor cells triggers the activation of the STING pathway (Supplementary Fig. 9a). Moreover, we observed that interferon-stimulated genes (ISGs) including *Ifnb1, Ifna2, Il-6,* and *Ccl5,* which have been shown to promote the recruitment of CD8$^+$ T lymphocytes into tumor sites[23–25], were significantly increased in *Arih1-WT*-OE tumor lysates (Fig. 3a, b), indicating that the activated STING pathway in *Arih1-WT*-OE tumors might be responsible for the establishment of CD8$^+$ T cell "inflamed" microenvironment.

The cytoplasmic double-stranded DNA (dsDNA) binds to the DNA sensor cGAS, which plays an essential role to promote STING pathway activation[26]. Therefore, we stained 4T1 cells with antibodies specific for dsDNA and cGAS proteins simultaneously. Multiple dsDNA foci in the cytoplasm, which were colocalized with cGAS proteins, were detected in *Arih1-WT*-OE 4T1 cells (Fig. 3c, Supplementary Fig. 9b), indicating that ARIH1 induced the accumulation of cytosolic dsDNA. Further, *Arih1-WT*-OE increased the levels of pSTING-S365 and pTBK1-S172 (Fig. 3d), and promoted nuclear translocation of IRF3 in 4T1 cells (Fig. 3e, Supplementary Fig. 9c), which are key molecular events following STING pathway activation[26]. Similar effect was also observed in human breast cancer cell lines MCF-7 and MDA-MB-231 (Fig. 3d). ARIH1-mediated STING pathway activation in 4T1 cells was mimicked using cisplatin treatment, and blocked by ARIH1 knockdown (Supplementary Fig. 9b, c). Notably, STING pathway activation, as indicated by the ISG gene expression, pSTING-S365 and pTBK1-S172 levels, was further enhanced in *Arih1-WT*-OE, but not *Arih1-C3SSS*-OE tumors treated with PD-L1 blockade (Supplementary Fig. 9d, e). Collectively, these results indicate that ARIH1-WT-OE promotes the activation of STING pathway.

We next tested whether STING pathway activation is required for ARIH1-WT-OE enhancing the efficacy of PD-L1 blockade. We knocked down *Sting* in *Arih1-WT*-OE 4T1 cell line (Fig. 4g), and found that mice bearing *Sting* knockdown *Arih1-WT*-OE tumors lost the sensitivity to PD-L1 blockade (Fig. 4a, b). Further, in *Sting* knockdown *Arih1-WT*-OE tumors, we did not observe any increase of CD8$^+$ or GzmB$^+$CD8$^+$ T cell infiltration, or any changes in the expression of ISG genes (*Ifnb1, Ifna2, Il-6,* and *Ccl5*) (Fig. 4c–f, h). In a second approach, we used C-176, which can covalently bind to Cys91 residue of STING protein and block its activation (Supplementary Fig. 10a)[27], to treat *Arih1-WT*-OE cells. Similar to *Sting*-knockdown, we observed that treatment with C-176 significantly abrogated the anti-tumor effects of PD-L1 blockade therapy in *Arih1-WT*-OE tumors, together with decreased CD8$^+$ and GzmB$^+$CD8$^+$ T cell infiltration, and reduced expression of ISG genes (Supplementary Fig. 10b–h). Together, these observations indicate that ARIH1-WT-OE promotes PD-L1 blockade-mediated anti-tumor immunity in a STING-dependent manner.

## ARIH1-mediated degradation of DNA-PKcs promotes STING pathway activation

To determine the key determinants for the ARIH1-WT-OE-mediated STING pathway activation, we performed a large-scale ARIH1 immunoprecipitation followed by mass spectrometry-based proteomic analysis to identify interacting proteins of ARIH1 (Fig. 5a). Notably, one of the top interacting proteins with ARIH1-WT was DNA-PKcs (DNA-dependent protein kinase catalytic subunit), which is a negative regulator of the STING pathway[28] (Fig. 5a). We also observed that the interaction of ARIH1 with DNA-PKcs was decreased in cells with the ligase-dead form of ARIH1 (ARIH1-C357S) (Fig. 5a, Supplementary Fig. 11a). The interaction between ARIH1 and DNA-PKcs was further confirmed with co-immunoprecipitation and GST pull-down assays (Fig. 5b, Supplementary Fig. 11b, c). Further, ARIH1 overexpression enhanced DNA-PKcs ubiquitination (Fig. 5c). Importantly, purified ARIH1 ubiquitinated purified DNA-PKcs in vitro, and this catalytic activity was blocked by C357S ligase-dead mutation of ARIH1 (Fig. 5d). Previous studies have shown that DNA-PKcs interacts with Ku70 and Ku80 to form a complex that mediates STING pathway inhibition[28, 29]. We found that the protein level of DNA-PKcs, but not of Ku70 or Ku80, decreased in ARIH1-WT cells, whereas no decrease was detected in ARIH1-C357S cells (Supplementary Fig. 11a), indicating that DNA-PKcs is a ubiquitination substrate of ARIH1.

Functionally, we discovered that ARIH1 overexpression in U2OS and HeLa cells resulted in the dose-dependent degradation of DNA-PKcs (Fig. 5e, Supplementary Fig. 11d), and ARIH1 knockdown resulted in the accumulation of DNA-PKcs (Fig. 5f, Supplementary Fig. 11e). We also observed that ARIH1 overexpression decreased DNA-PKcs protein levels in 4T1 tumors (Supplementary Fig. 11f). Further, the cycloheximide (CHX)-induced turnover of DNA-PKcs in HeLa cells was inhibited by knockdown of ARIH1 (Supplementary Fig. 11g). In contrast to a previously reported finding that degradation of DNA-PKcs depends on the ubiquitin-proteasome system[30], we observed that lysosomal inhibitors, including bafilomycin or $NH_4Cl$+Leup, but not the proteasome inhibitor MG132, restored DNA-PKcs protein levels in ARIH1-WT-OE cells (Fig. 5g). Immunofluorescence analysis showed that *Arih1-WT*-OE promoted the colocalization of DNA-PKcs with lysosomes (Fig. 5h). These results demonstrate that ARIH1 directly ubiquitinates DNA-PKcs and thereby marks it for lysosomal degradation.

It has been reported that inhibition of DNA-PKcs promotes cGAS-mediated STING pathway activation[28]. Therefore, we proposed that ARIH1-mediated degradation of DNA-PKcs may promote STING pathway activation. We initially aimed to define which domain in DNA-PKcs interacts with ARIH1. A series of flag-tagged DNA-PKcs truncated proteins were generated (Supplementary Fig. 11h). We found that only truncation protein G (G-flag) showed an exogenous ARIH1 interaction (Supplementary Fig. 11h). Next, we hypothesized that G-Flag competes with DNA-PKcs for binding to ARIH1, thereby reducing the degradation of DNA-PKcs by ARIH1 and further inhibiting the activation of the STING pathway. We found that this was indeed the case. Overexpression of G-Flag inhibited the interaction between ARIH1 and DNA-

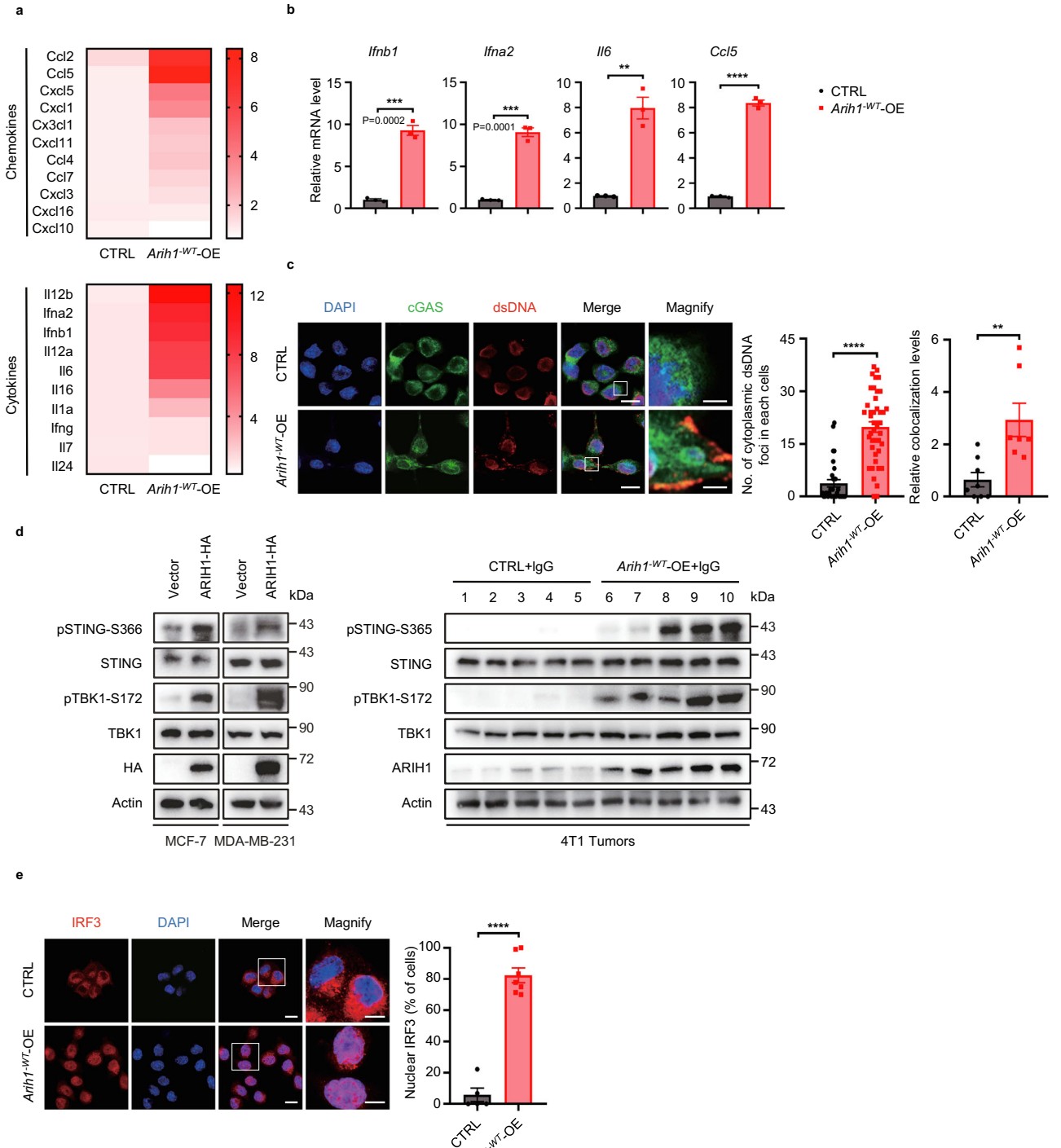

**Fig. 3 | ARIH1 promotes the STING pathway activation. a** Heatmap for the qRT-PCR expression data of genes for 10 cytokines and 11 chemokines in CTRL and *Arih1*-*WT*-OE 4T1 tumors (*n* = 3 per group). **b** qRT-PCR analysis for the gene expression of *Ifnb1*, *Ifna2*, *Il-6*, and *Ccl5* as in **a**. *n* = 3 per group. Data represent means ± SEM, \*\**P* < 0.01 (*P* = 0.0013), \*\*\**P* < 0.001, \*\*\*\**P* < 0.0001. **c** Immunofluorescence analysis of dsDNAs and cGAS in CTRL and *Arih1*-*WT*-OE 4T1 cells. The nuclei were stained with DAPI. Representative confocal images and quantitative data are shown. CTRL group (*n* = 31 cells), *Arih1*-*WT*-OE group (*n* = 43 cells). Scale bar, 10 μm; insets: Scale bar, 2 μm. Data represent means ± SEM, \*\**P* < 0.01 (*P* = 0.0042), \*\*\*\**P* < 0.0001. **d** Immunoblots of markers in the STING pathway including total and phospho

STING (S366 sites for human; S365 sites for mouse), total and phospho TBK1(S172) in lysates collected from breast cell lines (MCF-7 and MDA-MB-231) and 4T1 tumors (from Fig. 2a) with indicated treatments. **e** Immunofluorescent staining of IRF3 in CTRL and *Arih1*-*WT*-OE 4T1 cells and their quantifications. The nuclei were stained with DAPI. Scale bar, 10 μm; insets: Scale bar, 5 μm. Data represent means ± SEM, \*\*\*\**P* < 0.0001. Each dot in the graph represents the percentage of counted nucleus IRF3 cells in each sample, and the total number of counted cells in each group is as follows: CTRL group (*n* = 135 cells), *Arih1*-*WT*-OE group (*n* = 147 cells). For **b, c, e** data, Two-tailed *t*-test. Data shown in **c**–**e** are representative of three independent experiments. Source data are provided as a Source Data file.

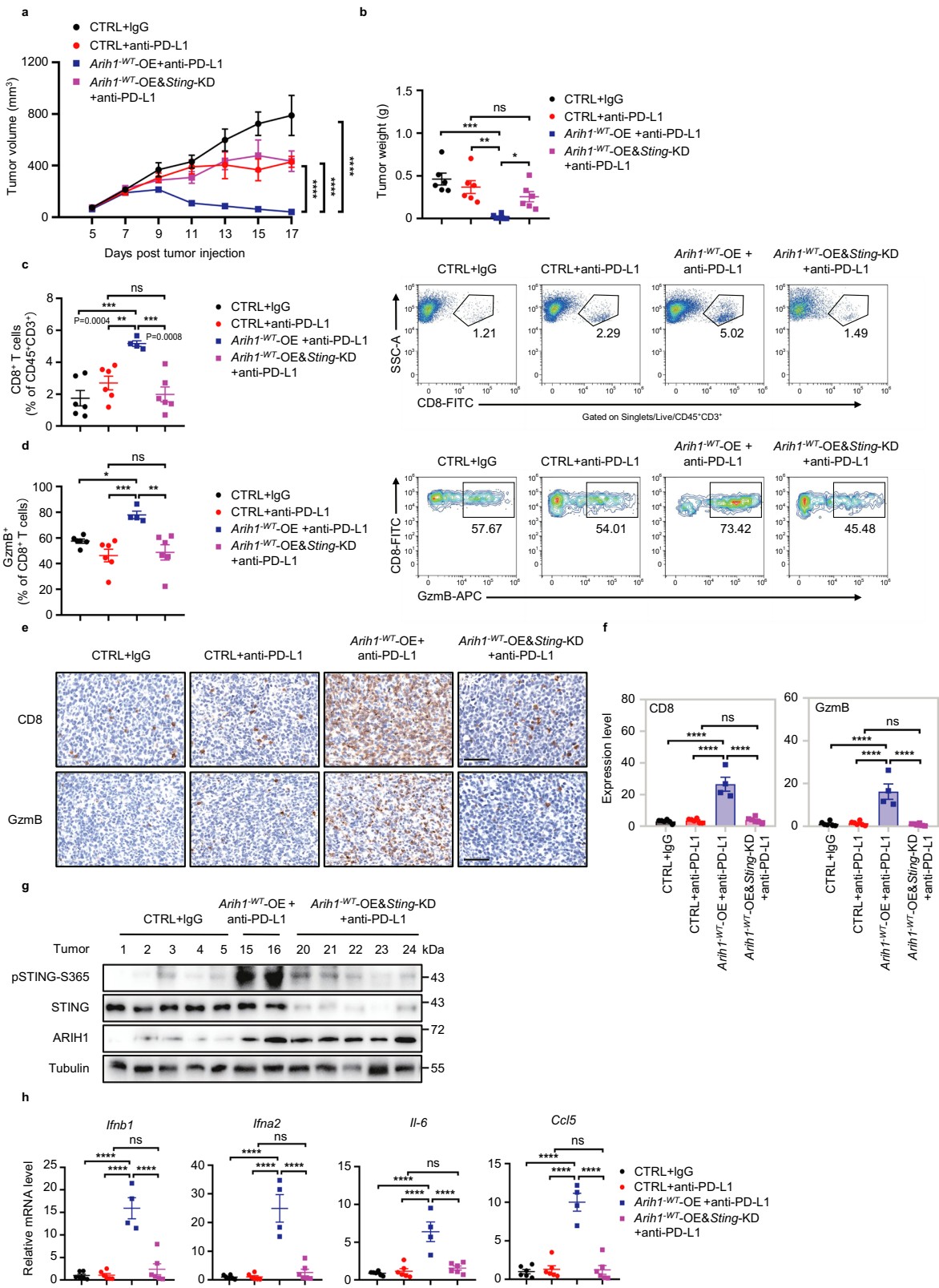

PKcs and the degradation of DNA-PKcs (Fig. 5i, Supplementary Fig. 11i) and suppressed the increase of pSTING-S366, pTBK1-S172 and ISG levels (Fig. 5i, Supplementary Fig. 11j), indicating that ARIH1-mediated activation of the STING pathway is dependent on the degradation of DNA-PKcs.

Further, we proposed that ARIH1 activates the STING pathway by promoting the degradation of DNA-PKcs in two ways: 1) promoting

dsDNA accumulation in the cytoplasm; and 2) reducing the inhibition of cGAS protein by DNA-PKcs[28]. We further explored which of these two mechanisms was predominant. Phosphorylation of cGAS by DNA-PKcs at T68 and S213 reduced its enzymatic activity, thus inhibit STING pathway activation[28]. We constructed the phospho-mimetic mutant T68E/S213D of cGAS to mimic the phosphorylation of cGAS by DNA-PKcs in cGAS knockdown MCF-7 and HeLa cells. Compared with WT

**Fig. 4 | Knockdown of STING reverses the anti-tumor effect of ARIH1- enhanced PD-L1 blockade therapy. a, b** Representative tumor growth curves and final tumor weights of tumors (including CTRL, *Arih1*[-WT]-OE, and *Arih1*[-WT]-OE&*Sting*-KD) in female BALB/c mice (6–8 week old) with the indicated treatments after subcutaneous injection of $5 \times 10^5$ 4T1 cells. $n = 6$ mice/group. Data represent means ± SEM. **a** ***$P < 0.0001$. **b** *$P < 0.05$ ($P = 0.0492$), **$P < 0.01$ ($P = 0.0027$), ***$P < 0.001$ ($P = 0.0002$), ns, not significant. **c, d** Representative figures and summary of frequency of tumor infiltrating CD8+ T cells and GzmB+CD8+ T cells of the mice as in **a**. $n = 6, 6, 4, 6$ mice/group. Data represent means ± SEM. **c** **$P < 0.01$ ($P = 0.0083$), ***$P < 0.001$, ns, not significant. **d** *$P < 0.05$ ($P = 0.0341$), **$P < 0.01$ ($P = 0.0022$),

***$P < 0.001$ ($P = 0.0010$), ns, not significant. **e, f** CD8 and GzmB IHC staining were performed in tumors of the mice as in **a**. The percent of each expression pattern was quantified **f**. Scale bar, 60 μm. $n = 6, 6, 4, 6$ mice/group. Data represent means ± SEM, ***$P < 0.0001$, ns, not significant. **g** Immunoblot analysis of pSTING-S365, STING, and ARIH1 in indicated tumor lysates for the experiment described in **a**. **h** qRT-PCR measurement of ISGs for the tumors of the mice as in **a**. $n = 6, 6, 4, 6$ mice/group. Data represent means ± SEM, ***$P < 0.0001$, ns, not significant. For **a** data, Two-way ANOVA test. For **b–d**, **f**, and **h** data, One-way ANOVA test. Source data are provided as a Source Data file.

cGAS, overexpression of ARIH1 in T68E/S213D cGAS cells failed to increase the phosphorylation levels of STING and TBK1, even though the levels of DNA-PKcs decreased and the levels of γ-H2AX (a specific marker for DNA damage) increased (Fig. 5j, Supplementary Fig. 11k). Additionally, T68E/S213D cGAS cells overexpressing ARIH1 also failed to induce ISGs expression compared to WT cGAS cells (Fig. 5k). These results indicate that dephosphorylation of T68/S213, the site where DNA-PKcs phosphorylates cGAS, is required for activation of the STING pathway even in the presence of DNA damage accumulation, further suggesting that the ARIH1-DNA-PKcs-cGAS axis is predominant for ARIH1-induced activation of the STING pathway.

To identify the relevance between DNA-PKcs and T cell infiltration in human tumors, we performed immunohistochemistry analysis of biopsies from human TNBC and normal tissues. We found that compared with normal tissues, tumor tissues expressed elevated protein levels of DNA-PKcs, together with low CD8 and ARIH1 (Supplementary Fig. 12b, c). We next found that normal tissues with high CD8 and ARIH1 levels showed prominent nuclear translocation of p-IRF3 (S396) (Supplementary Fig. 12d). Consistent results were found in specimens obtained from normal tissues and invasive breast carcinomas by analyzing the human tumor RNA-seq datasets from the TNMplot platform (Supplementary Fig. 12a). These results further imply that ARIH1 loss in human TNBC may mediate immune suppression and tumor progression similar to our findings in the mouse tumor models.

## The small molecule ACY738 enhances PD-L1 blockade therapy by increasing ARIH1 levels

Having demonstrated that ARIH1 can promote the antitumor effect of ICB, we next performed a luciferase-based high-throughput screening to identify small molecules for inducing ARIH1 levels. The screening model included HEK293T cells stably expressing Hibit-tagged ARIH1 and a library of 8207 drugs or drug candidates. We identified 5 hits that increase ARIH1 protein levels by>2.0-fold and show no effect on the proliferation of 4T1 cells (Fig. 6a). Among these molecules was ACY738, which is an HDACi (Histone Deacetylase inhibitors). ACY738 treatment led to both time-dependent and concentration-dependent induction of ARIH1 levels (Fig. 6b, Supplementary Fig. 13b) as well as phosphorylation of STING in 4T1 cells (Supplementary Fig. 13a), suggesting that ACY738 mediates induction of the ARIH1-STING axis.

To test whether ACY738 can induce anti-tumor immunity and enhance the efficacy of PD-L1 blockade, we treated mice bearing 4T1 tumors with ACY738 and/or anti-PD-L1. ACY738 monotherapy significantly reduced tumor growth, while anti-PD-L1 and ACY738 combination treatment led to a further reduction of tumor growth (Fig. 6c–e, Supplementary Fig. 13c). In addition, the combination treatment also significantly prolonged survival compared with anti-PD-L1 alone (Fig. 6f). In a lung metastatic 4T1 TNBC model, the number of tumor nodules was substantially reduced following combination treatment (Fig. 6g, h). Moreover, CD8+ T cells and GzmB+CD8+ T cells were significantly increased in tumors in the combination treatment group compared to anti-PD-L1 alone (Fig. 6i–k), together with increased levels of pSTING-S365, pTBK1-S172, and ARIH1 (Fig. 6l). Further, ACY738 treatment did not cause significant changes in body weight, survival, or pathological alterations in major organs in control

and ACY738-treated mice, whereas cisplatin treatment showed more pronounced toxicity (Supplementary Fig. 13d, Supplementary Fig. 16a–f).

Next, we aimed to determine whether the anti-tumor effect of ACY738-enhanced PD-L1 blockade therapy is dependent on ARIH1 or STING. We knocked down *Arih1* or *Sting* in 4T1 cells. We initially treated *Arih1* or *Sting* knockdown 4T1 cells with ACY738 in vitro, and observed no increase in the levels of pSTING-S365 or ISG genes (Supplementary Fig. 13e, f, Supplementary Fig. 14a, b). Further, the growth of the *Arih1* or *Sting* knockdown tumors was not altered upon combination treatment of ACY738 and PD-L1 blockade (Supplementary Fig. 13g, h, Supplementary Fig. 14c–e). Consistently, *Arih1* or *Sting* knockdown in tumors significantly abrogated the combination treatment-induced CD8+ T and GzmB+CD8+ cell infiltration (Supplementary Fig. 13i, Supplementary Fig. 14f, g) and activation of the STING pathway (Supplementary Fig. 13j). These data indicate that ACY738 exerts anti-tumor effects dependent on the ARIH1-STING axis. In addition, we observed that STING is also required for the synergy between cisplatin and PD-L1 checkpoint inhibition (Supplementary Fig. 14h–m).

Further, treatment of T68E/S213D cGAS cells with ACY738 did not increase the phosphorylation levels of STING and TBK1 compared to WT cGAS cells, even though the levels of DNA-PKcs decreased and the levels of ARIH1 and γ-H2AX increased (Supplementary Fig. 15a, b). Meanwhile, after ACY738 treatment, we observed a significant decrease in the levels of ISG genes in cells reconstituted with cGAS T68E/S213D compared to that reconstituted with cGAS WT (Supplementary Fig. 15c). These results indicate that ACY738 acts as an inducer of ARIH1, further confirming that ARIH1-mediated activation of the STING pathway occurs primarily through the ARIH1-DNA-PKcs-cGAS axis, rather than through the accumulation of DNA damage.

Collectively, our data indicate that ACY738 is a promising molecule and drug candidate to boost anti-tumor immunity and enhances PD-L1 blockade therapy by increasing the protein levels of ARIH1.

## Discussion

Primary or acquired resistance to ICB therapy in cancer patients, which results in low clinical response rates, remains a great therapeutic challenge. In this study, we observed that cisplatin enhanced the effect of PD-L1 blockade by resistant tumors by increasing ARIH1 protein levels. Moreover, genetic overexpression of Arih1 promoted resistant tumors more sensitive to PD-L1 blockade. In addition, we observed a positive correlation between ARIH1 expression and checkpoint blockade response in both mouse models and tumor patients. This sensitization was T cell-mediated and regulated by activating the intrinsic STING pathway following the ARIH1-induced degradation of DNA-PKcs. Tumor models revealed that ACY738, a identified inducer of ARIH1, enhanced the efficacy of PD-L1 blockade. These results suggest that ARIH1 can be a promising target for improving the clinical response to ICB therapy, thus providing a strategy for cancer treatment.

ARIH1 mutations are found in 1.4% of breast cancer samples ($n = 216$; deep deletion)[31], 2.9% of lung adenocarcinoma samples ($n = 35$; missense mutation)[32], 2.0% of prostate cancer samples ($n = 150$; missense mutation)[33] and 2.7% of melanoma samples ($n = 110$;

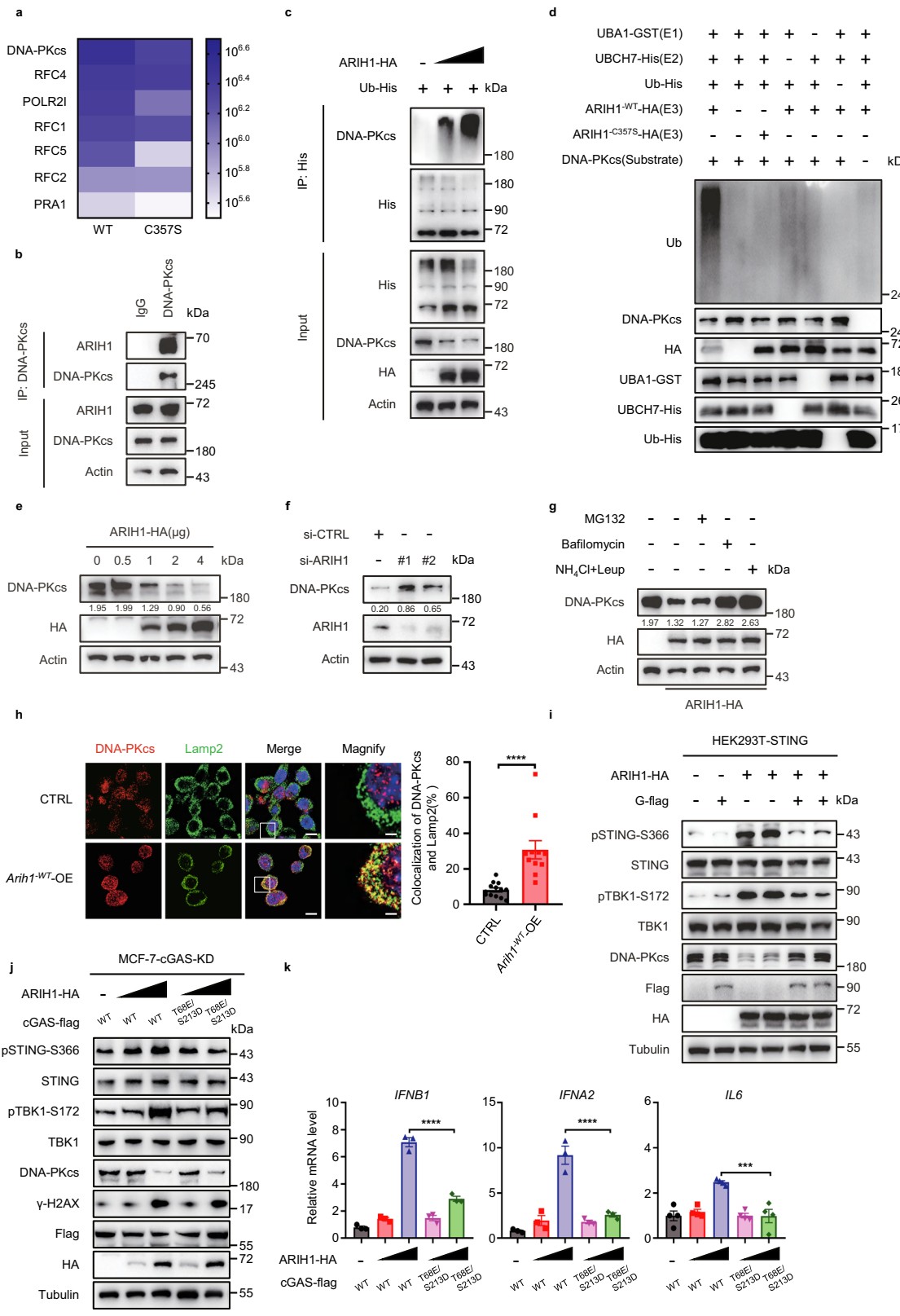

missense mutation)[34]. Interestingly, non-small cell lung cancer patients with the ARIH1 missense mutation (E429K) have reduced survival[35]. These intriguing mutations warrant further investigation in order to determine if they are relevant to anti-tumor immunity in different cancer types.

Previous cellular studies found that ARIH1, as a regulator of mitophagy, helped tumor cells maintain mitochondrial homeostasis

and overcome sensitivity to chemotherapy[15]. This suggests that ARIH1 is a tumor promoter in vitro. However, a more recent report has shown that ARIH1 catalyzed ISGylation of cGAS and thereby promoted antiviral immunity[36]. In combination with our findings, we found that ARIH1, as a tumor suppressor, participates in immune system-mediated tumor killing in vivo. In the absence of the immune system, ARIH1 may maintain normal tumor growth through mitophagy,

**Fig. 5 | ARIH1-mediated degradation of DNA-PKcs promotes STING pathway activation. a** HEK293T cells were transfected with ARIH1-WT-HA or ARIH1-C357S-HA for 36 h. Proteins that co-immunoprecipitated (Co-IP) with ARIH1 were analyzed by mass spectrometry. DNA repair-related proteins are shown in heatmap. **b** Co-IP of ARIH1 with DNA-PKcs in HEK293T cells. Endogenous DNA-PKcs was immunoprecipitated using anti-DNA-PKcs, and the immunoprecipitates were analyzed with anti-ARIH1. IgG, immunoglobulin G. **c** Immunoblot (IB) analysis for ubiquitination of DNA-PKcs from HEK293T cells co-transfected with the indicated constructs. **d** In vitro ubiquitination assay of purified DNA-PKcs. The reactions were performed with purified His-ubiquitin, GST-UBA1 (E1), His-UBCH7 (E2), and GST-ARIH1-WT or its ligase-dead mutant (GST-ARIH1-C357S) or in the absence of UBA1, UBCH7, ubiquitin, ARIH1 or DNA-PKcs. **e, f.** IB analysis of DNA-PKcs levels in U2OS cells. The cells were incubated with small interfering RNAs (siRNAs) against ARIH1 (**f**), or transfected with ARIH1-HA (**e**). **g** IB analysis of DNA-PKcs in U2OS cells transfected with the indicated constructs. Cells were treated by MG132, Bafilomycin or NH4Cl+Leup for 6 h. **h** Immunofluorescent staining of DNA-PKcs and Lamp2 in CTRL and Arih1-WT-OE 4T1 cells and their quantifications. The nuclei were stained with DAPI. Scale bar,

10 μm; insets: Scale bar, 2 μm. Data represent means ± SEM, ****$P < 0.0001$. Each dot in the graph represents the percentage of counted cells with co-localization in each sample, and the total number of counted cells in each group is as follows: CTRL group ($n = 381$ cells), Arih1-WT-OE group ($n = 321$ cells). **i** IB analysis of pSTING-S366, pTBK1-S172 and DNA-PKcs in HEK293T-STING cells co-transfected the indicated constructs. **j** cGAS-KD MCF-7 cells were transfected with cGAS WT or the phosphorylation-mimic mutants and ARIH1-HA. WCLs were analyzed by immunoblotting. **k** qRT-PCR measurement of ISGs expression in cGAS-KD HeLa cells transfecting with cGAS WT or the phosphorylation-mimic mutants and ARIH1-HA. IFNB1 ($n = 3$/group), IFNA2 ($n = 3$/group), IL6 ($n = 4$/group). Data represent means ± SEM, ***$P < 0.001$ ($P = 0.0002$), ****$P < 0.0001$. In **e–g**, the numbers under the blots represent the gray scale quantification (DNA-PKcs/Actin). For **k** data, One-way ANOVA test. For **h** data, Two-tailed t-test. Data shown in **b, c, e–h**, and **j, k** are representative of two independent experiments. Data shown in **a, d** and **i** are representative of three independent experiments. Source data are provided as a Source Data file.

but ARIH1 can also activate the STING pathway, which may not affect tumor growth. In the presence of the immune system, ARIH1 can alter the immune microenvironment of the tumor by stimulating the STING pathway, thereby activating immune system-mediated tumor killing. This is not contradictory to previous findings.

A number of small molecule agonists targeting STING have been developed. First-generation STING agonists were all cyclic dinucleotide candidates, and as such, it was not surprising that these agonists lacked stability and induced the production of inflammatory cytokines in normal tissues when systemically administered[24, 37, 38]. There is an urgent need to develop drugs or strategies to activate the STING pathway for clinical applications. Our study revealed that Arih1 overexpression in the 4T1 TNBC model, when combined with PD-L1 blockade, displayed complete tumor regression and led to a substantial survival benefit, which was mediated via STING activation. Importantly, ACY738, which activates ARIH1, also showed synergistic effects with anti-PD-L1 antibodies. These data suggest that more investigations are required to elucidate the therapeutic potential of ARIH1-enhancing drugs.

A previous study showed that DNA-PKcs activated the IRF3-mediated innate immune response in the presence of viral DNA[39]. In addition, DNA-PKcs is responsible for STING-independent innate immune activation in human cells[40]. Alternatively, a recent study revealed that cGAS phosphorylation was suppressed by cytoplasmic DNA-PKcs, thereby inhibiting STING signaling activation[28]. Consistent with this result, we identified that DNA-PKcs mRNA levels negatively correlated with gene signatures of the STING pathway in a large number of tumors from TCGA datasets (Supplementary Fig. 17). Furthermore, we observed that blocking ARIH1-induced degradation of DNA-PKcs restored its protein level and inhibited STING pathway activation. Since DNA-PKcs is a key protein in the DNA damage repair[41], this may be a possible reason for the degradation of DNA-PKcs by ARIH1 leading to the accumulation of dsDNA in the cytoplasm.

It is not surprising, as ARIH1 can degrade DNA-PKcs, we further proposed that ARIH1 may affect DNA damage repair. We used antibodies specific for γ-H2AX and the tumor protein p53 binding protein 1 (53BP1), which are specific markers of the DNA damage response (DDR). Immunofluorescence and Western blot analysis showed that the levels of γ-H2AX significantly decreased (Supplementary Fig. 18a, c, e), while 53BP1 levels increased in cisplatin-treated ARIH1 knockdown 4T1 and U2OS cells (Supplementary Fig. 18b, d), suggesting that DDR is enhanced after knockdown of ARIH1. Additionally, ARIH1 knockdown also reduced the accumulation of dsDNA foci in cisplatin-treated 4T1 cells (Supplementary Fig. 9b). These data suggest that ARIH1 can affect cisplatin-induced DNA damage. However, we observed that the DNA damage caused by

ARIH1 overexpression is significantly less than that caused by cisplatin treatment (Supplementary Fig. 9b), so our study provides a safer therapeutic strategy targeting ARIH1 compared to cisplatin treatment.

In summary, our study sheds light on the mechanism of overcoming resistance to checkpoint blockade that is regulated by the upregulation of ARIH1 and its associated T cell-mediated effects. These effects are driven by activation of the STING pathway via ARIH1-DNA-PKcs-cGAS axis. Our work suggests that ARIH1 may be a target to induce STING activation, and more investigations are needed to elucidate the therapeutic potential of ARIH1 activators.

## Methods
### Reagents and antibodies
The following compounds were purchased from Topscience (Shanghai, China): Cisplatin (#T1564), C-176 (#T5154), and ACY738 (#T3509). MG-132 (#S2619), Bafilomycin A1 (#S1413), leupeptin Hemisulfate (#S7380) were from Selleck. NH4Cl (A501569) was purchased from Sangon®Biotech (Shanghai, China). The antibodies were provided as follows: DNA-PKcs (#ET1610-12, 1: 2000, HUABIO), DNA-PKcs (#sc-5282, 1:200, Santa Cruz Biotechnology), TBK1 (#3504, 1: 1000, Cell Signaling Technology), p-TBK1 (Ser172) (#AP1026, 1:1000, ABclonal), IRF3 (#A0816, 1:100, ABclonal), p-IRF3 (Ser396) (#29047, 1:100, Cell Signaling Technology), STING (#13647, 1:1000, Cell Signaling Technology), STING (#ET1705-68, 1:1000, HUABIO), p-STING (Ser366) (#19781, 1:1000, Cell Signaling Technology), p-STING (Ser366) (#AP1223, 1:1000, ABclonal), cGAS (#HA500023, 1: 1000, HUABIO), Lamp2 (#sc-18822, 1:200, Santa Cruz Biotechnology), dsDNA (#sc-58749, 1:200, Santa Cruz Biotechnology), ARIH1 (C-7) (#sc-514551, 1: 50; Santa Cruz Biotechnology), ARIH1 (Goat) (#EB05812, 1:1000, Everestbiotech), GST (B-14) (#sc-138, 1: 200, Santa Cruz Biotechnology), γ-H2AX (#ET1602-2, 1: 1000, HUABIO), H2AX (#ET1705-97, 1: 1000, HUABIO), 53BP1 (#ET1704-05, 1: 1000, HUABIO), Ubiquitin (P4D1) (#sc-8017, 1: 200, Santa Cruz Biotechnology), His-tag (#66005-1-Ig, 1: 1000, Proteintech), Flag-tag (#0912-1, 1: 2000, HUABIO), HA-tag (#0906-1, 1: 2000, HUABIO), Tubulin (#M1305-2, 1:5000, HUABIO) and β-Actin (#M1210-2, 1: 2000, HUABIO). The secondary antibodies for western blot were used: goat anti-mouse (#31430,1:20000, Thermo Fisher Scientific), goat anti-rabbit (#31460, 1:20000, Thermo Fisher Scientific), donkey anti-goat (#A0181, 1:1000, Beyotime). The fluorescent secondary antibodies for immunofluorescence were used: goat anti-rabbit Alexa Fluor 555 (#A-21428, 1:500, Thermo Fisher Scientific), goat anti-mouse DyLight 649 (#A23610, 1:500, Abbkine). The following beads were used for immunoprecipitation: Anti-Flag (DYKDDDDK) Affinity Gel (#B23102) and Anti-HA magnetic beads (#B26202) were purchased from Bimake, Protein A/G agarose beads (#B23201) was

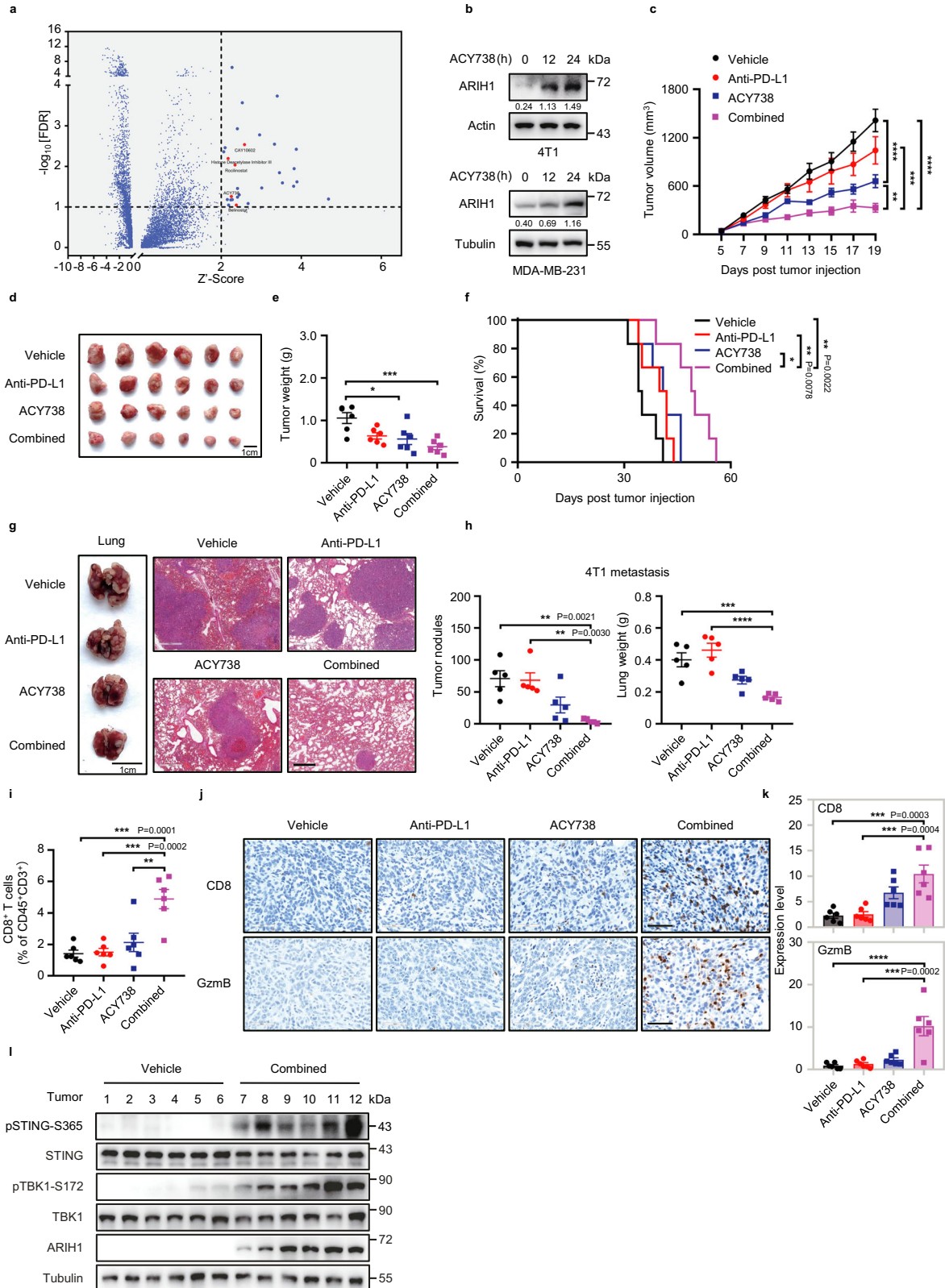

purchased from Cell Signaling Technology. The antibodies for immunohistochemistry (IHC) were used: ARIH1 (#EB05812, 1:100, Everestbiotech), CD8α (#98941, 1:200, Cell Signaling Technology), CD8α (#ab17147, 1:100, Abcam), GzmB (#44153, 1:50, Cell Signaling Technology), IRF3 (#A11118, 1:100, ABclonal), DNA-PKcs (#ET1610-12, 1:200, HUABIO). The following antibodies for flow analysis were displayed: Zombie Violet™ Fixable Viability Kit (#423114; 1:200; Biolegend),

PerCP/Cyanine5.5 anti-mouse CD45 (#103132; 1:200; Biolegend), PE/Cyanine7 anti-mouse CD3 (#100320; 1:200; Biolegend), FITC anti-mouse CD8 (#100706; 1:200; Biolegend), APC anti-human/mouse Granzyme B (#372204; 1: 200; Biolegend). The in vivo antibodies for mouse models were used: control antibody InVivoMab rat IgG2b isotype (#BE0090; 100 or 200 μg; Bioxcell), anti-PD-L1 (#BE0101; 100 or 200 μg; Bioxcell).

**Fig. 6 | The small molecule ACY738 increases ARIH1 protein levels and enhances PD-L1 blockade therapy in 4T1 tumor models. a** High-throughput screening of 8207 drug or drug candidates was performed to screen for ARIH1 enhancers that increase ARIH1 levels. After treatment with the drugs at 5 μM for 24 h, luciferase as a reporter to detect ARIH1 protein levels following exposing to the substrate (10 μM) in 384-well plates. Hit compounds are shown as red dots. **b** Immunoblots (IB) analysis for the ARIH1 levels in MDA-MB-231 and 4T1 cells treated with 1 μM ACY738 at the indicated time points. The numbers under the blots represent the gray scale quantification (ARIH1/Actin and ARIH1/Tubulin). **c** Tumor growth curves from subcutaneous injection of $5 \times 10^5$ 4T1 cells in the female BALB/c mice (6–8 week old) treated with vehicle, anti-PD-L1 alone, ACY738 alone, and ACY738+anti-PD-L1. $n = 6$ mice/group. Data represent means ± SEM, $^{**}P < 0.01$ ($P = 0.0012$), $^{***}P < 0.001$ ($P = 0.0005$), $^{****}P < 0.0001$. **d–f.** Representative image, final tumor weights, and survival curves of tumor bearing mice with the indicated treatments. $n = 6$ mice/group. Data represent means ± SEM, **e** $^*P < 0.05$ ($P = 0.0151$),

$^{***}P < 0.001$ ($P = 0.0010$). Log-rank test (**f**), $^*P < 0.05$ ($P = 0.0200$), $^{**}P < 0.01$. **g, h** Representative images and quantification of spontaneous lung metastases and lung weights of BALB/c mice (6–8 week old) at day 16 with the indicated treatments after intravenous injection of $1 \times 10^5$ 4T1 cells. Scale bar, 400 μm. $n = 5$ mice/group. Data represent means ± SEM, $^{**}P < 0.01$, $^{***}P < 0.001$ ($P = 0.0008$), $^{****}P < 0.0001$. **i** Quantification of tumor-infiltrating CD8$^+$ T cells of the mice as in **c**. $n = 6$ mice/group. Data represent means ± SEM, $^{**}P < 0.01$ ($P = 0.0018$), $^{***}P < 0.001$. **j, k** The CD8 and GzmB IHC staining (**j**) was performed in tumors of the mice as in **c**. The percent of each expression pattern was quantified (**k**). Scale bar, 60 μm. $n = 6$ mice/group. Data represent means ± SEM, $^{***}P < 0.001$, $^{****}P < 0.0001$. **l** IB analysis of total and phospho STING (S365), total and phospho TBK1(S172), and ARIH1 in tumor lysates of the mice as in **c**. For **c** data, Two-way ANOVA test. For **e, h-i**, and **k** data, One-way ANOVA test. Data shown in **a, b** and **i** are representative of three independent experiments. Source data are provided as a Source Data file.

## Cell culture

HEK293T (ATCC CRL-3216), HeLa (ATCC CRM-CCL-2), U2OS (ATCC HTB-96), MCF-7 (ATCC HTB-22), MDA-MB-231 (ATCC CRM-HTB-26), 4T1 (ATCC CRL-2539), E0771 (ATCC CRL-3405), B16-F10 (ATCC CRL-6475), LLC (ATCC CRL-1642), MC38 (Kerafast ENH204-FP), and CT26 (ATCC CRL-2638) cell lines were obtained from the American Type Culture Collection (ATCC) and Kerafast, Inc.. HEK293T, HeLa, U2OS, MCF-7, MDA-MB-231, E0771, B16-F10, MC38, and LLC cell lines were cultured in DMEM (Hyclone, with L-glutamine, with 4.5 g/L glucose, without pyruvate) containing 10% FBS (Gibco™) and 1% antibiotics (streptomycin and penicillin, Gibco™) at 37 °C with 5%CO$_2$ atmosphere. 4T1 and CT26 cell lines were maintained in RPMI-1640 (Hyclone, with L-glutamine) supplemented with 10% FBS and 1% penicillin/streptomycin. 4T1-*Arih1*$^{WT}$-OE, 4T1-*Arih1*$^{C3SSS}$-OE, 4T1-*Arih1*$^{WT}$-OE&*Sting*-KD, 4T1-*Arih1*-KD, 4T1-*Sting*-KD, E0771-*Arih1*$^{WT}$-OE, E0771-*Arih1*$^{C3SSS}$-OE, B16-F10-*Arih1*$^{WT}$-OE, and B16-F10-*Arih1*$^{C3SSS}$-OE cells were generated by our laboratory through lentiviral transduction.

Overexpressing *Arih1* in murine tumor cells (including 4T1, E0771 and B16-F10) was generated by lentiviral infection. To generate lentiviruses, PCDH-CTRL, PCDH-murine *Arih1*$^{WT}$, or PCDH-murine *Arih1*$^{C3SSS}$ with two helper plasmids psPAX2 and pMD2.G were co-transfected into HEK293T cells and viral supernatant was harvested 48 and 72 h post-transfection. After being filtered through a 0.45 μm filter to remove live cells, the virus was collected with a concentrated solution (25% PEG8000, 0.75 M NaCl) at 4 °C overnight. Collected lentiviruses were used to infect cells with polybrene (5 μg/mL) before centrifugation at 3000 *g* for 20 min. The stable single cell lines were generated with puromycin (2 μg/mL or 4 μg/mL) for 3 days and then isolated by cell sorting (Moflo-Mstrios EQ, Beckman). The ARIH1 levels were detected by qRT-PCR and western blot.

## shRNA and siRNA-mediated knockdown

The lentiviral shRNA targets murine genes using multiple shRNA constructs that have been cloned into the pLV3 or pLKO.1 vector. Lentiviruses were packaged in HEK293T cells according to described above. 4T1 or 4T1-*Arih1*$^{WT}$-OE cells were transfected with condensed virus particles for 24 h and selected by puromycin. Knockdown efficiency was assessed by qRT-PCR and western blot. siRNAs were transfected into HeLa, U2OS, and MCF-7 cells using Lipofectamine™ 2000 (Invitrogen™), according to manufacturer's protocol. The siRNA and shRNA target sequences in this study are provided in Supplementary Table 1.

## Cell proliferation assay

Cells expressing CTRL and *Arih1*$^{WT}$-OE were seeded in 96-well cell culture plates (Corning 3603™) at a density of 3000 cells per well in 100 μL DMEM medium supplemented with 10% FBS. The cell number at the indicated time points was determined by a cell viability assay (CellCounting-Lite® 2.0 Luminescent Cell Viability Assay, #DD1101-03)

and measuring the luminescence value with the Varioskan™ LUX instrument (ThermoFisher). Each sample was repeated five times.

## In vitro high-throughput drug screening

The full-length ARIH1 CDS region with its promoter and Hibit (Nano-Glo®) were cloned into plasmid HP138 using recombination. The recombinant plasmid and helper plasmid HP216 were co-transfected into HEK293T cells, followed by puromycin treatment to select cells with Hibit-tagged ARIH1 integrating into its genome. For high-throughput drug screening, drug screening cells were transfected with Lgbit plasmid for 24 h, and about 5000 cells were plated into each well in 50 μL DMEM medium to 384 well plates (Corning 3764™). After the cells had grown for 12 h, the final concentration of 5 μM tested compounds were added through the pipetting workstation, followed by culturing for another 24 h. Then, furimazine (at a final concentration 10 μM) was added to each well to measure the luminescent reading values in cytation5. Each drug was repeated three times.

Relative luminosity values of each drug were calculated using the Luminous read value of the drug well divided by that of the DMSO control. Z'-score was calculated using relative luminosity values with its expected to be 1. FDR was calculated using the Benjamini-Hochberg method for each drug screening plate. The hit compounds were picked and classified according to the Z'-score>2, FDR < 0.1, and cell viability.

## Immunoprecipitation and immunoblotting

Cells were cultured and transfected plasmids for 36 h, as described above. Whole-cell extracts were lysed in RIPA buffer [20 mM Tris-HCl (pH 7.5), 150 mM NaCl, 0.5% NP-40, 1 mM NaF, 1 mM Na3VO4, 1 mM EDTA (pH 7.5)] supplemented with Protease and Phosphatase Inhibitor Cocktail (bimake) on ice for 1 h. The supernatants were collected after centrifuging at 12,000 × *g* for 10 min at 4 °C and incubated with coupled agarose beads at 4 °C overnight. After washing three times with RIRA buffer, beads were collected and subjected to immunoblotting. For immunoblotting analysis, protein samples were heated at 100 °C for 10 min, subjected to SDS-PAGE, and then transferred onto PVDF membranes. After blocking with PBST buffer containing 5% (w/v) skimmed milk at room temperature for 1 h, membranes were probed with the indicated primary antibodies at 4 °C overnight. Following incubation with corresponding secondary antibodies at room temperature for 1 h and subsequent washing, blots were detected using chemiluminescence reagents (#4AW001-500, 4A649 Biotech, Co.). Images were captured using a Gel imaging system (Tanon 4600).

## Mass spectrometry and data analysis

For mass spectrometry analysis of the binding partners of ARIH1, the proteins obtained by immunoprecipitation against overexpressed ARIH1$^{WT}$-HA or ARIH1$^{C357S}$-HA in HEK293T cells were trypsin digested on beads. For details, the immunoprecipitated proteins were resolved in 8 M urea and 500 mM Tris–HCl (pH 8.5). Disulfide bridges were

reduced by adding Tris (2-carboxyethyl) phosphine (TCEP) at a final concentration of 5 mM for 20 min. Reduced cysteine residues were then alkylated by adding 10 mM iodoacetamide (IAA) and incubating for 15 min in the dark at room temperature. The urea concentration was reduced to 2 M by adding 100 mM Tris−HCl (pH 8.5) and 1 mM CaCl2. The protein mixture was digested overnight at 37 °C with trypsin at an enzyme-to-substrate ratio of 1:100 (w/w). The resulting peptides were analyzed on a Thermo Scientific Q Exactive HF-X mass spectrometer.

The protein identification and quantification were done by Max-Quant v1.5[42]. For details, the tandem mass spectra were searched against the UniProt human protein database and the built-in contaminant protein list. Trypsin was set as the enzyme, and the specificity was set to both N and C terminal of the peptides. The maximum missed cleavage was set to 2. The cysteine carbamidomethylation was set as a static modification, and the methionine oxidation was set as a variable modification. The precursor and fragment mass tolerance were set as 20 ppm. The first-search peptide mass tolerance and main-search peptide tolerance were set to 20 and 4.5 ppm, respectively. The false discovery rate at the peptide spectrum match level and protein level was controlled to be <1%. Only unique peptides and razor peptides were used for quantification, and the minimum ratio count for protein identification was 2. The summed peptide intensities were used for protein quantification.

## Protein purification and In vitro ubiquitination assays

His-tagged protein UBCH7 (E2) and ubiquitin production was induced by 1 mM IPTG to E. coli BL21. The tagged proteins were purified using Ni$^+$-NTA affinity column followed by elution with lysis buffer (PBS with 300 mM imidazole). Protein concentrations were determined using a Bradford Protein Assay (Bio-Rad). Plasmids GST-UBA1 (E1), HA-ARIH1 (E3), and ARIH1 inactive mutant (C357S) were transfected into HEK293T cells for 36 h and lysed in RIPA Lysis Buffer followed by the addition of concentrated proteins using anti-GST or anti-HA agarose beads. Endogenous DNA-PKcs was collected using protein A/G agarose beads for 2 h at 25 °C before incubation anti-DNA-PKcs primary antibody overnight at 4 °C. An in vitro ubiquitination assay was carried out as below. Briefly, reactions were performed in a 40 μL reaction mixture at 37 °C for 2 h in the presence of His-Ub, E1, E2, E3, DNA-PKcs, ATP regeneration solution (#BML-EW9810-0100, Enzo Life Sciences) and Ubiquitin Buffer (#BML-KW9885-0005, Enzo Life Sciences). All reactions were terminated by boiling 10 min with 2x SDS sample buffer and then subjected to western blot.

## Immunofluorescence imaging

Cells were cultured in 12-well plates on glass coverslips, washed twice with PBS, and then fixed in 4% paraformaldehyde at room temperature for 20 min. Cells were permeabilized in blocking buffer (PBS containing 5% FBS, 0.1% Triton X-100) for 1 h at 25 °C and incubated with primary antibodies overnight at 4 °C in a humidified chamber. After washing twice with PBS, coverslips were incubated with secondary antibodies for 1 h at 25 °C and imaged with Zeiss LSM 880 with AiryScan. ImageJ software was used to process graphics.

## RNA extraction and RT-qPCR analysis for tumor cytokines

Total RNA was extracted using TRIzol™ Plus RNA Purification Kit (Invitrogen) according to the manufacturer's protocol. Extracted RNA (1 μg) was transcribed into cDNA using ChamQ Universal SYBR qPCR Master Mix according to the manufacturer's instructions (Vazyme). HiScript® II Q RT SuperMix (Vazyme) and gene-specific primers (sequences listed in Supplementary Table 2) were performed for quantitative real-time PCR by using the Step One Plus Real-Time PCR Systems (ABI). β-Actin was used as an internal control.

## RNA-Seq and pathway enrichment analysis

Total RNA was isolated from Arih1$^{WT}$-OE and Arih1$^{C355S}$-OE 4T1 cells as described above. The samples were then sent to Shanghai Majorbio Bio-Pharm Technology Co., Ltd for transcriptome sequencing through the Illumina NovaSeq 6000 sequencer. RNA quality control was uniformly performed.

Pathway enrichment was performed with Gene set enrichment analysis (GSEA). Briefly, log2Foldchange in differential expression analysis of comparisons was provided as input to the GSEA analysis though using the R package clusterProfiler (version 4.2.2). The KEGG gene sets file was downloaded from https://www.kegg.jp. P value cutoff of 0.05 and an absolute value of normalized enrichment score cutoff of 1 were used in selecting significant gene sets.

## Immunostaining

Tumor and normal tissues were collected and processed from six female patients (6 cases each group, median age: 51, range from 43 to 59) with triple-negative breast cancer at Taizhou Hospital of Zhejiang Province affiliated to Wenzhou Medical University, following ethical guidelines. All patients willingly provided their samples at no cost after signing the informed consent form. The selection of cancer patients was random, and no biases related to sex or gender were performed.

For immunohistochemistry of human TNBC tumors and normal samples, tissues were rapidly fixed with 4% paraformaldehyde and embedded in paraffin for tissue sections (4 μm thick). Then, sections were incubated with the following primary antibodies: anti-ARIH1 (1:100), anti-CD8α (1:100), and anti-DNA-PKcs (1:200). For immunofluorescent staining, sections were stained with the following primary antibodies: anti-p-IRF3 (S396) (1:100). And for immunohistochemistry of murine 4T1 tumor samples, the primary antibodies used are anti-ARIH1 (1:100), anti-CD8α (1:200) and anti-GzmB (1:50). Visualization was done using the Olympus BX61 light microscope and ScanScope CS2. The staining intensity and percentage of positive cells were analyzed and used to generate an H-score for each sample that passed quality control.

## Ethical approval

All mice were kept in a specific pathogen-free (SPF) facility. Carbon dioxide was used for euthanasia. All the animal experiments were strictly conducted in accordance with the protocols approved by the Tab of Animal Experimental Ethical Inspection of the First Affiliated Hospital, College of Medicine, Zhejiang University (Reference Number: (2022) Real Action Fast Review No. (847)).

## Mouse tumor models

All mice of 6–8 weeks of age were purchased from Shanghai SLAC Laboratory Animal, Co., Ltd (Shanghai, China). All mice were cultured in suitable temperature and humidity environment (25 °C, suitable humidity (typically 50%), 12 h dark/light cycle), and fed with sufficient water and food.

To establish syngeneic mouse tumor models, 4T1 TNBC cells (5 × 10$^5$) were injected subcutaneously in 50 μL PBS and Matrigel (1: 1 v/v) into the second breast fat pad of female BALB/c or nude mice. E0771 cells (8 × 10$^5$) were suspended in 50 μL PBS and Matrigel (1: 1 v/v) and were injected subcutaneously into the second breast fat pad of female C57BL/6 mice. Suspensions of B16-F10 cells (2.5 × 10$^5$ or 5 × 10$^5$) in 100 μL PBS were subcutaneously implanted in the dorsal flank of C57BL/6 male mice. On days 3-5 after injection, tumor size was measured and calculated by using the formula 1/2 × length × width$^2$. Mice with similar tumor burdens were randomized into treatment groups. We used 4T1 and E0771 cells in female mice to mimic human triple-negative breast cancer (TNBC). 4T1 and E0771 cells can be transplanted into the fat pad of the mouse mammary gland, in contrast to male mice, these tumor cells are highly tumorigenic, invasive, and

spontaneously metastatic. Experiments with 4T1 and E0771 cells in female mice could provide a valuable model system for preclinical TNBC studies. For the tumor metastasis model involving the B16-F10 cell line, we selected male mice according to a previous study that male mice are more suitable for studying melanoma tumor models within the immune microenvironment[43].

Cisplatin (#T1564, Topscience) and ACY738 (#T3509, Topscience) were dissolved in saline. For treatment, tumor-bearing mice were randomly divided into 4 groups and given: 1) control antibody InVivoMab rat IgG2b isotype (#BE0090; Bioxcell); 2) cisplatin (5 mg/kg, 1 out of 7 days) or ACY738 (5 mg/kg, daily); 3) anti-PD-L1 (#BE0101; Bioxcell, 100 or 200 μg, 3 out of 7 days); and 4) anti-PD-L1 plus cisplatin or anti-PD-L1 plus ACY738 by intraperitoneal injection. Cisplatin-treated mice received additional saline supplementation (0.5 ml daily x 2 days following cisplatin by intraperitoneal injection). The cisplatin dose and frequency chosen was the weekly tolerated dose that did not have severe side effects on mice[44]. To inhibit STING, mice were injected intraperitoneally with 750 nmol C-176 per mouse in 200 μl corn oil (solarbio) three times a week for a total 2-week treatment course[27]. No adverse reactions were observed during the treatment. Mice were sacrificed when tumor volumes exceeded 2000 mm$^3$ or when tumors reached over 20 mm in any dimension. For survival studies, mice were monitored and measured for tumor volumes once a week after their initial injection.

### Tumor sample preparation and flow cytometry

Tumors were collected and processed into single-cell suspensions through digestion in collagenase type I (#2350118, Gibco) and Dnase I (#143582, Roche) at 37 °C for 45 min. After filtering with a 45 μm filter (BD Bioscience), the isolated cells were stained with the specific surface marker antibodies, anti-CD45-Percp-Cy5.5 (#103132; Biolegend), anti-CD3-PE-Cy7 (#100320; Biolegend) and anti-CD8-FITC (#100706; Biolegend) in PBS for 30 min at 4 °C. Intracellular staining of GzmB was performed as follows: cells were washed and then fixed and permeabilized with a Fix/Perm kit (#421403; Biolegend) and finally stained with anti-APC-GzmB (#372204; Biolegend). For proper compensation of flow cytometry channels, single-stain samples were utilized. The stained cells were analyzed on the flow cytometer (Beckman Coulter Cytoflex), and data were analyzed using CytExpert2.4 software.

### Statistics and reproducibility

Numerical data are presented as mean ± SEM; all statistical data analyses were performed with GraphPad Prism 8.0 software. Two-tailed t-test (where two groups of data were compared) or One-way ANOVA (where more than two groups of data were compared) was used to analyze the statistical differences with $P$-values indicated in the related graphs. For animal studies, Two-way ANOVA test was used to determine statistical significance for time points when all mice were viable for tumor measurement. Kaplan–Meier survival curves and corresponding log-rank (Mantel-Cox) tests were used to evaluate the statistical differences between groups in survival studies. A significant difference exists when $P < 0.05$. All assays were carried out at least two or three independent times with the same results.

### Reporting summary

Further information on research design is available in the Nature Portfolio Reporting Summary linked to this article.

## Data availability

The mass spectrometry proteomics data generated in this study have been deposited to the ProteomeXchange Consortium via the iProX partner repository with the dataset identifier PXD041429. The RNA-seq data from *Arih1$^{WT}$*-OE and *Arih1$^{-C3SSS}$*-OE 4T1 cells generated in this study have been deposited in the Gene Expression Omnibus (GEO) database under the accession numbers GSE231726. The human cancer data

(Supplementary Fig. 1a, Supplementary Fig. 8a-c, Supplementary Fig. 12a, and Supplementary Fig. 17) were derived from databases listed in Supplementary Table 3. The remaining data are available within the article, Supplementary information, and source data file. Source data are provided with this paper.

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

## Acknowledgements

Financial support from the National Natural Science Foundation of China (32222023 and 32100598 to H.X.) and the Natural Science Foundation of Zhejiang Province (LR22C070002 to X.C.) is gratefully acknowledged. We thank Prof. Hui Yang from Institute of Neuroscience, Chinese Academy of Sciences for pcDNA3-STING-flag and pcDNA3-cGAS-flag plasmids, Prof. Teng Ma from Beijing Institute of Radiation Medicine for different DNA-PKcs truncation mutants, Prof. Shanshan Pei, Jiaming Chen and Haoxin Xu from Zhejiang University Medical Center for their helpful suggestions to this project.

## Author contributions

H.X. conceived and coordinated the project. H.X. designed the experiments. H.X. interpreted the data and wrote the manuscript. X.L., X.C., R.W., Z.C. and Y.X. performed most of the experiments. F.W., B.S., L.Z., J.Z., B.X., Y.C., J.H., Y.L., Y.W., C.Z., and D.W. assisted with the experiments and helped to analyze the data.

## Competing interests

The authors declare no competing interests.
