## [Peer Review File · Nature Communications]

ARIH1 activates STING-mediated T-cell activation and sensitizes tumors to immune checkpoint blockadeREVIEWER COMMENTS

Reviewer #1 (Remarks to the Author): with expertise in DNA-PK, cGAS/STING

In this manuscript, the authors found that cisplatin treatment enhanced the antitumor effect of PD-L1 blockade. Next, authors showed that ARIH1 was induced by cisplatin treatment and was required for the antitumor effect of cisplatin+anti-PL-L1. Then authors then revealed that ARIH1 promoted cGAS-STING signaling and knockdown of STING ablated the antitumor activity of anti-PD-L1 towards ARIH1-expressing tumors. Mechanistically, the authors identified DNA-PKcs, a negative regulator of STING signaling, as a substrate of ARIH1 and showed that ARIH1 expression led to the lysosomal degradation of DNA-PKcs. Finally, authors did a compound library screen and identified a compound ATCY738 that could promote ARIH1 expression level and showed synergistic antitumor effect when combined with ICI treatment. Overall, the whole study is well designed, the data are extensive and most data support the conclusion. This reviewer has a few concerns listed below:

1. Figure 3a, 3b, authors showed that ARIH1 expression induced interferon and cytokine production, probably due to the engagement of STING signaling. Authors also mentioned that ARIH1 expression did not affect tumor growth in immunodeficient mice (line 128-129). This reviewer was curious whether ARIH1 expression inhibited cell growth in vitro, since type I interferons are known to suppress cell proliferation.
2. Authors showed that ARIH1 led to DNA-PKcs degradation, which is convincing. Then authors relied on DNA-PKcs downregulation to explain the enhanced STING signaling, which is interesting, but to this reviewer, the link is not obvious. Figure 5i is a critical piece of data, and this reviewer got what the authors wanted to convey in the main manuscript. However, the reviewer would strongly suggest that the authors rephrase the writings to make things clear (line 240-245). Furthermore, it could be more convincing if authors could show that the mutant they generated functions in a dominant negative manner to block the interaction between ARIH1 and DNA-PKcs.
3. The authors demonstrated the specificity of ACY738 by showing no antitumor effect in the treatment of ARIH1 KD tumors. However, this reviewer would suggest that ACY738 be tested in cell culture, which is more straightforward. Based on the authors' theory, ACY738 treatment could be able to enhance basal STING signaling and further promote interferon signaling upon STING activation. Moreover, the activity of ACY738 on interferon signaling should be dependent on ARIH1 as well as STING.

Reviewer #2 (Remarks to the Author): with expertise in cancer biology, drug discovery

The authors present a strong series of experiments identifying ARIH1-mediated activation of the STING pathway as a trigger for enhanced sensitivity to checkpoint inhibitors in tumors. The data are of high quality and warrant the conclusions of this manuscript. Throughout the manuscript the potential involvement of DNA accumulation in the recruitment and/or activation of the T cell response is not addressed. Some additional experiments to rule out this effect would further strengthen the manuscript.

Fig 1. The impact of cisplatin is major. The additional effect of pd-1 inhibition seems marginal especially in 1b,c but more dramatic in fig 1h-j. The upregulation of ARIH1 in response to cisplatin is very clear. Altogether, data in this figure could be explained by enhanced cisplatin-induced DNA damage or altered DNA damage response in presence of ARIH1. Accumulation of cells with DNA damage might subsequently lead to T cell infiltration and further tumor killing. Is the level of DNA damage or activation of DDR in presence of cisplatin affected by ARIH1 knockdown?

Fig 2. Convincingly shows that overexpression of ARIH1 renders 4T1 tumors sensitive to anti-PD-L1 even without cisplatin and shifts the tumor from cold to hot. The growth of ARIH1 overexpressing tumors without any treatment is severely impacted. This may point to accumulation of DNA damage already without cisplatin. However, Extended Data Figure 4 using nude mice convincingly rules out tumor growth arrest in absence of T cells. The use of the ubiq-dead mutant convincingly shows

ARIH1 ubiq function is needed to render tumors sensitive to enhanced T cell infiltration.

Extended Data Figure 5. The figures show that ARIH1 expression positively correlates with response to checkpoint expression in patients. The results in fig1 suggest that induction of DNA damage is required to trigger ARIH1 expression in 4T1 cells. In patients this seems not to be the case. Is the strict requirement for chemotherapy to induce ARIH1 observed only in the 4T1 model? Does a model naturally expressing ARIH1 show a similar selection as seen in patients?

Fig 3 and extended data fig 6. Induction of STING signaling by ARIH1 overexpression is clear and is also seen in other cell lines. The cisplatin experiments combined with ARIH1 knockdown further indicate STING activation may underlie the results shown in fig1.

Fig 4 and extended data fig 7. Strong data showing that STING depletion or inhibition essentially attenuates all effects mediated by ARIH1 overexpression, clearly showing that STING activation is needed for ARIH1-mediated activation of the anti-tumor immune response. Is STING required in a similar manner for the synergy between cisplatin and checkpoint inhibition?

Fig 5 and extended data figs 8,9. DNA-PK is identified as a substrate for ARIH1 and impact on STING activation is shown. The finding that elevated levels of DNA-PKcs are accompanied by reduced CD8 and ARIH1 levels in tumor tissues confirms the model. DNA-PK also cooperates with ATR and ATM to control the DNA damage response. Degradation of DNA-PK may impact DNA repair. Again, can the authors rule out involvement of DNA damage accumulation as a mechanism driving T cell recruitment/ activation?

Fig 6 and extended data fig 10. A small molecule screen identifies the HDACi ACY738 as an inducer of ARIH1 that mimics the effect of cisplatin in an ARIH1 dependent manner. Again, can the authors rule out involvement of DNA damage accumulation in response to the induced epigenetic changes, as a mechanism driving T cell recruitment/ activation?

Reviewer #3 (Remarks to the Author): with expertise in cancer immunology/immunotherapy

Summary:

Breast cancer is typically insensitive to ICB therapy. The authors find that cisplatin treatment can sensitize 4T1 breast cancer to ICB therapy and implicate ARIH1 induction. KD of ARIH1 induction is sufficient to sensitize tumors to ICB in a STING-dependent manner. Drug screen identified ACY738 as a potent inducer of ARIH that can sensitize 4T1 to ICB and at the doses utilized in mice, is better tolerated than cisplatin.

Overall, with caveats noted below, the authors have done a complete job in investigating ARIH1 a tumor-intrinsic STING-dependent mediator of ICB efficacy. This points to a potentially clinically actionable target to enhance tumor immunogenicity. Whether this applies to other tumor models and the overlap with other reported mechanisms of immunogenic cell death will be the subject of future investigations.

Major points:

Figure 1-2: The data are compelling and clear. However, for the key in vivo expts (A, H), there is no mention of how many times an expt was performed. In general, it is best practice to repeat in vivo tumor growth experiments with 3 total experiments.

Figure 6: Does ACY738 work in the Sting KD tumors?

Minor points:

Line 88: Authors write on line 88 that they used "4T1-derived murine TNBC xenografts". This is likely a typo since 4T1 into BALB/c is syngeneic. Xenografts is also written throughout the figure legends. If

the data are derived from true xenografts, then all of these experiments would be problematic since immunotherapy treatment in humans cannot be modeled in settings of xenogeneic mismatch.

Line 89: No corrections requested, but for what it is worth, other groups have shown sensitivity of 4T1 to PD-1 blockade. See Zappasodi, Merghoub et al. Nat Medicine 2019.

Fig 4E: "CRTL" is not spelled correctly.

Reviewer #1 (Remarks to the Author): with expertise in DNA-PK, cGAS/STING

In this manuscript, the authors found that cisplatin treatment enhanced the antitumor effect of PD-L1 blockade. Next, authors showed that ARIH1 was induced by cisplatin treatment and was required for the antitumor effect of cisplatin+anti-PL-L1. Then authors then revealed that ARIH1 promoted cGAS-STING signaling and knockdown of STING ablated the antitumor activity of anti-PD-L1 towards ARIH1-expressing tumors. Mechanistically, the authors identified DNA-PKcs, a negative regulator of STING signaling, as a substrate of ARIH1 and showed that ARIH1 expression led to the lysosomal degradation of DNA-PKcs. Finally, authors did a compound library screen and identified a compound ATCY738 that could promoted ARIH1 expression level and showed synergistic antitumor effect when combined with ICI treatment. Overall, the whole study is well designed, the data are extensive and most data support the conclusion. This reviewer has a few concerns listed below:

Response:

We thank the reviewer for their appreciation of our study and the constructive comments. Following the suggestions, we have conducted additional experiments and added a total of 50 new experimental panels to the manuscript. The newly added content is highlighted in red text. We hope that our response sufficiently addresses the concerns raised by the reviewer. Please note that the figure citations in our response below refer to the new (post-revision) figures.

1. Figure 3a, 3b, authors showed that ARIH1 expression induced interferon and cytokine production, probably due to the engagement of STING signaling. Authors also mentioned that ARIH1 expression did not affect tumor growth in immunodeficient mice (line 128-129). This reviewer was curious whether ARIH1 expression inhibited cell grow *in vitro*, since type I interferons are known to suppress cell proliferation.

Response:

We thank the reviewer for this suggestion. We have now performed cell viability assays *in vitro* to assess the proliferation of control (CTRL) and Arih1-overexpressing (Arih1^{-WT}-OE) cells, and included the results in Extended Data Figure 5a and below. The result showed that the growth of 4T1 cells was not affected by Arih1 overexpression *in vitro*. Similar phenotypes were observed in two other models, including E0771 and B16-F10. IFN signaling can inhibit tumor growth by promoting the activation and infiltration of immune cells [PMID:36154696]. Due to the absence of immune cells in *in vitro* cultures, Arih1 overexpression cannot inhibit the growth of tumor cells *in vitro* even though it induces IFN expression. Although Arih1 overexpression did not affect the proliferation of tumor cells *in vitro*, the growth of Arih1^{-WT}-OE tumor cells was greatly diminished in wild-type mice (Fig. 2a, b), but not in immunodeficient nude mice (Extended Data Fig. 5b, c), as compared to control tumor cells *in vivo*, supporting our proposed model of ARIH1-elicited T cell-mediated antitumor immunity via the DNA-PKcs-STING signaling.

Overexpression of ARIH1 shows no effect on the proliferation of tumor cells *in vitro*.

Extended Data Figure 5a: Tumor cells were infected with an empty vector (CTRL) or Arih1 overexpressing lentiviral preparations (Arih1^{WT}-OE). Cell viability was monitored at indicated time points by an ATP assay. n=5/group. For a data is presented as mean ± SEM, Two-way ANOVA test. ns, not significant.

Overexpression of ARIH1 significantly inhibits tumor growth in immunocompetent mice.

Figure 2a-b: a. Tumor growth curves in BALB/c mice with control (CTRL) and Arih1-overexpressing (Arih1^{WT}-OE) tumors treated with PD-L1 or isotype mAbs intraperitoneally (i.p.) starting on day 7 and then every three days after subcutaneous inoculation of 5×10^5 4T1 cells. n=6 mice/group. b. Representative image of tumors of tumor bearing mice at Day25 with the indicated treatments. n=6 mice/group. Data shown in a is representative of three independent experiments. For a data is presented as mean ± SEM, Two-way ANOVA test. ****P < 0.0001.

Overexpression of ARIH1 shows no effect on the tumor growth in immunocompromised mice.

2. Authors showed that ARIH1 led to DNA-PKcs degradation, which is convincing. Then authors relied on DNA-PKcs downregulation to explain the enhanced STING signaling, which is interesting, but to this reviewer, the link is not obvious. Figure 5i is a critical piece of data, and this reviewer got what the authors wanted to convey in the main manuscript. However, the reviewer would strongly suggest that the authors rephrase the writings to make things clear (line 240-245). Furthermore, it could be more convincing if authors could show that the mutant they generated functions in a dominant negative manner to block the interaction between ARIH1 and DNA-PKcs.

Response:

We thank the reviewer for this comment. We have performed co-immunoprecipitation (Co-IP) assays to test whether the mutant (G-Flag) can indeed block the interaction between ARIH1 and DNA-PKcs and included the results in Extended Data Figure 11i and below. Our findings showed that overexpression of G-flag in HEK293T-STING cells expressing ARIH1-HA resulted in the accumulation of DNA-PKcs levels and blocked the interaction between ARIH1 and DNA-PKcs, suggesting that G-flag acts as a dominant-negative inhibitor in our model.

In addition, we have now amended the text (line 240-245) accordingly as below.

Next, we hypothesized that G-Flag competes with DNA-PKcs for binding to ARIH1, thereby reducing the degradation of DNA-PKcs by ARIH1 and further inhibiting the activation of the STING pathway. We found that this was indeed the case. Overexpression of G-Flag inhibited the interaction between ARIH1 and DNA-PKcs and the degradation of DNA-PKcs (Fig. 5i, Extended Data Fig. 11i) and suppressed the increase of pSTING-S366, pTBK1-S172 and ISG levels (Fig. 5i, Extended Data Fig. 11j), indicating that ARIH1-mediated activation of the STING pathway is dependent on the degradation of DNA-PKcs.

Overexpression of G-flag in HEK293T-STING cells expressing ARIH1-HA resulted in accumulation of DNA-PKcs levels and weakened DNA-PKcs and ARIH1 interactions.

Extended Data Figure 11i: Co-IP of ARIH1 with DNA-PKcs or its truncated mutant (G-Flag) in HEK293T-STING cells. Exogenous ARIH1 was immunoprecipitated using anti-HA, and the immunoprecipitates were analyzed with anti-DNA-PKcs and anti-Flag.

3. The authors demonstrated the specificity of ACY738 by showing no antitumor effect in the treatment of ARIH1 KD tumors. However, this reviewer would suggest that ACY738 be tested in cell culture, which is more straightforward. Based on the authors' theory, ACY738 treatment could be able to enhance basal STING signaling and further promote interferon signaling upon STING activation. Moreover, the activity of ACY738 on interferon signaling should be dependent on ARIH1 as well as STING.

Response:

We thank the reviewer for this suggestion. We treated ARIH1-KD or STING-KD 4T1 cells with ACY738 *in vitro* and assessed the activation of the STING pathway by western blot and qRT-PCR and added the data to Extended Data Figure 13e-f and Extended Data Figure 14a-b and below. We observed a significant increase in phosphorylation of STING and expression of ISG genes in Ctrl-KD 4T1 cells after ACY738 treatment, while ARIH1-KD or STING-KD 4T1 cells failed to induce phosphorylation of STING and expression of ISG genes, supporting our proposed mechanism of ACY738-induced anti-tumor immunity via the ARIH1-STING axis.

Activation of the STING pathway by ACY738 is dependent on ARIH1 and STING.

Extended Data Figure 13e-f: **e.** Immunoblots analysis of pSTING-S366 and ARIH1 in Ctrl-KD and Arih1-KD 4T1 cells treated with ACY738 1μM for 24 hours. **f.** qRT-PCR measurement of ISGs expression in Ctrl-KD compared to Arih1-KD 4T1 cells with indicated treatments. n=3-6/group. For **f** data is presented as mean ± SEM, One-way ANOVA test. **P < 0.01, ***P < 0.001, ****P < 0.0001, ns, not significant.

Extended Data Figure 14a-b: **a.** Immunoblots analysis of STING and ARIH1 in Ctrl-KD and Sting-KD 4T1 cells treated with ACY738 1μM for 24 hours. **b.** qRT-PCR measurement of ISGs expression in Ctrl-KD compared to Sting-KD 4T1 cells with indicated treatments. n=3-6/group. For **b** data is presented as mean ± SEM, One-way ANOVA test. *P < 0.05, **P < 0.01, ns, not significant.

Reviewer #2 (Remarks to the Author): with expertise in cancer biology, drug discovery

The authors present a strong series of experiments identifying ARIH1-mediated activation of the STING pathway as a trigger for enhanced sensitivity to checkpoint inhibitors in tumors. The data are of high quality and warrant the conclusions of this manuscript. Throughout the manuscript the potential involvement of DNA accumulation in the recruitment and/or activation of the T cell response is not addressed. Some additional experiments to rule out this effect would further strengthen the manuscript.

Response:

We thank the reviewer for their appreciation of our study and the constructive

comments. Following the suggestions, we have conducted additional experiments and added a total of 50 new experimental panels to the manuscript. The newly added content is highlighted in red text. We hope that our response sufficiently addresses the concerns raised by the reviewer. Please note that the figure citations in our response below refer to the new (post-revision) figures.

Fig 1. The impact of cisplatin is major. The additional effect of pd-11 inhibition seems marginal especially in 1b,c but more dramatic in fig 1h-j. The upregulation of ARIH1 in response to cisplatin is very clear. Altogether, data in this figure could be explained by enhanced cisplatin-induced DNA damage or altered DNA damage response in presence of ARIH1. Accumulation of cells with DNA damage might subsequently lead to T cell infiltration and further tumor killing. Is the level of DNA damage or activation of DDR in presence of cisplatin affected by ARIH1 knockdown?

Response:

We thank the reviewer for this comment. We have now performed a series of experiments and added the data to Extended Data Figure 18a-e and below.

In this experiment, we used antibodies specific for phosphorylated histone H2AX (γ -H2AX) and tumor protein p53 binding protein 1 (53BP1), which are specific markers of the DNA damage response (DDR). Immunofluorescence and Western blot analysis showed that the levels of γ -H2AX significantly decreased (Extended Data Fig. 18a, c, e), while 53BP1 levels increased in cisplatin-treated ARIH1 knockdown 4T1 and U2OS cells (Extended Data Fig. 18b, d), suggesting that DDR is enhanced after knockdown of ARIH1. Additionally, ARIH1 knockdown also reduced the accumulation of dsDNA foci in cisplatin-treated 4T1 cells (Extended Data Fig. 9b). These data suggest that ARIH1 can affect cisplatin-induced DNA damage. It is not surprising, as ARIH1 can degrade DNA-PKcs, which is a key protein regulating DNA damage repair [PMID: 33424929]. Therefore, when ARIH1 is knocked down, DNA-PKcs-mediated damage repair is accelerated and DNA damage is inhibited.

However, we observed that the DNA damage caused by ARIH1 overexpression is significantly less than that caused by cisplatin treatment (Extended Data Fig. 9b), so our study provides a safer therapeutic strategy targeting ARIH1 compared to cisplatin treatment.

It has been reported that inhibition of DNA-PKcs promotes cGAS-mediated STING pathway activation [PMID: 33273464]. We also found that DNA-PKcs knockdown significantly increased the formation of γ -H2AX foci and cytosolic micronuclei (formed by dsDNA breaks) in U2OS cells (data not shown). Consequently, we proposed that ARIH1 activates the STING pathway by promoting degradation of DNA-PKcs in two ways: 1) promoting dsDNA accumulation in the cytoplasm; and 2) reducing the inhibition of cGAS protein by DNA-PKcs [PMID: 33273464].

We further explored which of these two mechanisms was predominant.

Phosphorylation of cGAS by DNA-PKcs at T68 and S213 reduced its enzymatic activity, thus inhibit STING pathway activation [PMID: 33273464]. We constructed the phospho-mimetic mutant T68E/S213D of cGAS to mimic the phosphorylation of cGAS by DNA-PKcs in cGAS knockdown MCF-7 and HeLa cells. Compared with WT cGAS, overexpression of ARIH1 in T68E/S213D cGAS cells failed to increase the phosphorylation levels of STING and TBK1, even though the levels of DNA-PKcs decreased and the levels of γ -H2AX increased (Fig. 5j, Extended Data Fig. 11k). Additionally, T68E/S213D cGAS cells overexpressing ARIH1 also failed to induce ISGs expression compared to WT cGAS cells (Fig. 5k). These results indicate that even in the presence of DNA damage accumulation, dephosphorylation of T68/S213, the site where DNA-PKcs phosphorylates cGAS, is required for activation of the STING pathway, further suggesting that the ARIH1-DNA-PKcs-cGAS axis is predominant for ARIH1-induced activation of the STING pathway (see response to comment #4).

ARIH1 can affect cisplatin-induced DNA damage.

Extended Data Figure 18a-b: Immunofluorescence analysis of γ -H2AX (**a**) and 53BP1 (**b**) in Ctrl-KD and Arih1-KD 4T1 cells at the indicated times following cisplatin treatment. The nuclei were stained with DAPI (blue). A representative immunofluorescence image is shown. More than 400 cells (**a**) or 30 cells (**b**) were analyzed per group. Scale bar, 10 μ m. For **a-b** data are presented as means \pm SEM, unpaired Student's t-test. *P < 0.05, **P < 0.01, ****P < 0.0001, ns, not significant.

Extended Data Figure 18c-d: Immunofluorescence analysis of γ -H2AX (**c**) and 53BP1 (**d**) in U2OS cells transfected with ARIH1-siRNA at the indicated times following cisplatin treatment. The nuclei were stained with DAPI (blue). A representative immunofluorescence image is shown. More than 100 cells (**c**) or 30 cells (**d**) were analyzed per group. Scale bar, 10 μ m. For **c-d** data are presented as means \pm SEM, unpaired Student's t-test. *P < 0.05, **P < 0.01, ns, not significant.

Extended Data Figure 18e: Immunoblots analysis of γ -H2AX and ARIH1 levels in 4T1 and U2OS cells after treatment with 10 μ M cisplatin for indicated times.

Extended Data Figure 9b: Immunofluorescence analysis of dsDNAs (red) and cGAS (green) in CTRL, Arih1^{-WT}-OE and Arih1-KD 4T1 cells with indicated treatments. The nuclei were stained with DAPI (blue). A representative immunofluorescence image is shown. More than 30 cells were analyzed per group. Scale bar, 10 μ m. For **b** data is presented as mean \pm SEM, One-way ANOVA test. **P < 0.01, ***P < 0.001, ****P < 0.0001.

The phospho-mimetic mutant T68E/S213D of cGAS inhibits the activation of STING pathway mediated by ARIH1 degradation of DNA-PKcs.

Figure 5j-k: **j.** cGAS-KD MCF-7 cells were transfected with cGAS WT or the phosphorylation-mimic mutants and ARIH1-HA. WCLs were analyzed by immunoblotting. **k.** qRT-PCR measurement of ISGs expression in cGAS-KD HeLa cells transfecting with cGAS WT or the phosphorylation-mimic mutants and ARIH1-HA. n=3-4/group. For **j-k**, two independent experiments are conducted. For **k** data is presented as mean \pm SEM, One-way ANOVA test. ***P < 0.001, ****P < 0.0001.

Fig 2. Convincingly shows that overexpression of ARIH1 renders 4T1 tumors sensitive to anti-PD-L1 even without cisplatin and shifts the tumor from cold to hot. The growth of ARIH1 overexpressing tumors without any treatment is severely impacted. This may point to accumulation of DNA damage already without cisplatin. However, Extended Data Figure 4 using nude mice convincingly rules out tumor growth arrest in absence of T cells. The use of the ubiq-dead mutant convincingly shows ARIH1 ubiq function is needed to render tumors sensitive to enhanced T cell infiltration.

Extended Data Figure 5. The figures show that ARIH1 expression positively correlates with response to checkpoint expression in patients. The results in fig1 suggest that induction of DNA damage is required to trigger ARIH1 expression in 4T1 cells. In patients this seems not to be the case. Is the strict requirement for chemotherapy to induce ARIH1 observed only in the 4T1 model? Does a model naturally expressing ARIH1 show a similar selection as seen in patients?

Response:

1) Is the strict requirement for chemotherapy to induce ARIH1 observed only in the 4T1 model?

We thank the reviewer for the comment. We have now investigated whether cisplatin increases ARIH1 expression in additional tumor models and added the data to Extended Data Figure 3 and below. Cisplatin treatment resulted in significantly increased levels of ARIH1 *in vitro* across various cancer models, including 4T1 and E0771 breast cancer, B16-F10 melanoma, LLC lung cancer, as well as MC38 and CT26 colon cancer models.

Cisplatin increases ARIH1 protein levels in various mouse cancer cell lines.

Extended Data Figure 3: Immunoblot analysis of ARIH1 levels in tumor cells after treatment with 10 μ M cisplatin for indicated times. The numbers under the blots represent the gray scale quantification (ARIH1/Tubulin).

2) Does a model naturally expressing ARIH1 show a similar selection as seen in patients?

We thank the reviewer for the comment. Given that the existing data (Extended Data Figure 8) indicate a positive correlation between natural ARIH1 expression and checkpoint blockade response in patients, it is crucial to determine if mouse models with naturally expressed ARIH1 exhibit similar phenotypes to patients. We have now conducted a series of experiments and added the data to Extended Data Figure 7a-k and below.

We compared ARIH1 protein levels in six murine cancer cell lines and found that B16-F10 cells had relatively high levels of ARIH1 (ARIH1-High), while 4T1 had relatively low levels of ARIH1 (ARIH1-Low) (Extended Data Fig. 7a). Subsequently, we tested the response of two types of tumor cells to PD-L1 blockade in syngeneic mouse models. Compared to 4T1 tumors (ARIH1-Low), treatment of B16-F10 tumors (ARIH1-High) with PD-L1 blockade greatly reduced tumor growth and resulted in a significant survival benefit, consistent with the sensitivity to PD-L1 checkpoint blockade observed clinically in patients with advanced melanoma [PMID: 33476492, PMID: 36460017] (Extended Data Fig. 7b, c). In addition, we performed single-cell sorting of the same type of tumor cells (4T1 and E0771) and selected monoclonal cells with relatively high and low ARIH1 protein levels, respectively, to test their response to PD-L1 blockade. We found that, in the same type of tumor cells, mice with tumors with relatively high ARIH1 expression (ARIH1-High) were more sensitive

to PD-L1 treatment, with substantially reduced tumor growth and higher tumor inhibition rate compared to the group with relatively low ARIH1 expression (ARIH1-Low) (Extended Data Fig. 7d-k). These results suggest a positive correlation between ARIH1 expression and tumor response to checkpoint blockade therapy in mouse models, consistent with the patient data shown in Extended Data Figure 8.

ARIH1 expression positively correlated with tumor response to checkpoint blockade therapy in mouse models.

Extended Data Figure 7a-c: **a.** Immunoblot analysis of ARIH1 levels in tumor cells. The numbers under the blots represent the gray scale quantification (ARIH1/Tubulin). **b.** Tumor growth curves (left and middle) and survival curves (right) of B16-F10 cells (2.5×10^5) in C57BL/6 mice ($n=5-12$ per group) with indicated treatments. **c.** Tumor growth curves (left and middle) and survival curves (right) of 4T1 cells (5×10^5) in BALB/c mice ($n=7$ per group) with indicated treatments. Two-way ANOVA test (data are presented as means \pm SEM) was used to determine statistical significance for time points when all mice were viable for tumor measurement. Log-rank (Mantel-Cox) test was used to determine the statistical significance for the survival of mice. ***P < 0.001, ****P < 0.0001, ns, not significant.

Fig 3 and extended data fig 6. Induction of STING signaling by ARIH1 overexpression is clear and is also seen in other cell lines. The cisplatin experiments combined with ARIH1 knockdown further indicate STING activation may underlie the results shown

in fig1.

Fig 4 and extended data fig 7. Strong data showing that STING depletion or inhibition essentially attenuates all effects mediated by ARIH1 overexpression, clearly showing that STING activation is needed for ARIH1-mediated activation of the anti-tumor immune response. Is STING required in a similar manner for the synergy between cisplatin and checkpoint inhibition?

Response:

We thank the reviewer for this comment. We have now performed STING knockdown in the 4T1 tumor model with cisplatin-PD-L1 inhibitor combination and added the data to Extended Data Figure 14h-m and below.

With Sting knockdown, tumors continued to progress even when treated with the combined cisplatin and PD-L1 blockade therapy (Extended Data Fig. 14h-k). Consistently, tumors with Sting knockdown did not show an increase in CD8⁺ T cell or GzmB⁺CD8⁺ T cell infiltration following combination therapy (Extended Data Fig. 14l-m). These results show that STING is required for the synergy between cisplatin and PD-L1 checkpoint inhibition.

data are presented as means \pm SEM, One-way ANOVA test. For **k** data is presented as mean \pm SEM, unpaired Student's t-test. *P < 0.05, ***P < 0.001, ****P < 0.0001, ns, not significant.

Fig 5 and extended data figs 8,9. DNA-PK is identified as a substrate for ARIH1 and impact on STING activation is shown. The finding that elevated levels of DNA-PKcs are accompanied by reduced CD8 and ARIH1 levels in tumor tissues confirms the model. DNA-PK also cooperates with ATR and ATM to control the DNA damage response. Degradation of DNA-PK may impact DNA repair. Again, can the authors rule out involvement of DNA damage accumulation as a mechanism driving T cell recruitment/ activation?

Response:

We thank the reviewer for this comment. We have now performed a series of experiments and added the data to Figure 5j-k and Extended Data Figure 11k and below.

It has been reported that inhibition of DNA-PKcs promotes cGAS-mediated STING pathway activation [PMID: 33273464]. We also found that DNA-PKcs knockdown significantly increased the formation of γ -H2AX foci and cytosolic micronuclei (formed by dsDNA breaks) in U2OS cells (data not shown). Consequently, we proposed that ARIH1 activates the STING pathway by promoting degradation of DNA-PKcs in two ways: 1) promoting dsDNA accumulation in the cytoplasm; and 2) reducing the inhibition of cGAS protein by DNA-PKcs [PMID: 33273464]. We further explored which of these two mechanisms was predominant.

Phosphorylation of cGAS by DNA-PKcs at T68 and S213 reduced its enzymatic activity, thus inhibit STING pathway activation [PMID: 33273464]. We constructed the phospho-mimetic mutant T68E/S213D of cGAS to mimic the phosphorylation of cGAS by DNA-PKcs in cGAS knockdown MCF-7 and HeLa cells. Compared with WT cGAS, overexpression of ARIH1 in T68E/S213D cGAS cells failed to increase the phosphorylation levels of STING and TBK1, even though the levels of DNA-PKcs decreased and the levels of γ -H2AX increased (Fig. 5j, Extended Data Fig. 11k). Additionally, T68E/S213D cGAS cells overexpressing ARIH1 also failed to induce ISGs expression compared to WT cGAS cells (Fig. 5k). These results indicate that even in the presence of DNA damage accumulation, dephosphorylation of T68/S213, the site where DNA-PKcs phosphorylates cGAS, is required for activation of the STING pathway, further suggesting that the ARIH1-DNA-PKcs-cGAS axis is predominant for ARIH1-induced activation of the STING pathway.

The phospho-mimetic mutant T68E/S213D of cGAS inhibits the activation of STING pathway mediated by ARIH1 degradation of DNA-PKcs.

Figure 5j-k: **j.** cGAS-KD MCF-7 cells were transfected with cGAS WT or the phosphorylation-mimic mutants and ARIH1-HA. WCLs were analyzed by immunoblotting. **k.** qRT-PCR measurement of ISGs expression in cGAS-KD HeLa cells transfected with cGAS WT or the phosphorylation-mimic mutants and ARIH1-HA. n=3-4/group. For **j-k**, two independent experiments are conducted. For **k** data is presented as mean \pm SEM, One-way ANOVA test. ***P < 0.001, ****P < 0.0001.

Extended Data Figure 11k: cGAS-KD HeLa cells were transfected with cGAS WT or the phosphorylation-mimic mutants and ARIH1-HA. WCLs were analyzed by immunoblotting. For **k**, two independent experiments are conducted.

Fig 6 and extended data fig 10. A small molecule screen identifies the HDACi ACY738 as an inducer of ARIH1 that mimics the effect of cisplatin in an ARIH1 dependent manner. Again, can the authors rule out involvement of DNA damage accumulation in

response to the induced epigenetic changes, as a mechanism driving T cell recruitment/ activation?

Response:

We thank the reviewer for this comment. We have now performed a series of experiments and added the data to Extended Data Figure 15a-c and below.

We reconstituted the phospho-mimetic mutants T68E/S213D of cGAS to mimic the phosphorylation of cGAS by DNA-PKcs in cGAS knockdown MCF-7 and HeLa cells. Compared to WT cGAS, treatment of T68E/S213D cGAS cells with ACY738 did not increase the phosphorylation levels of STING and TBK1, even though the levels of DNA-PKcs decreased and the levels of ARIH1 and γ -H2AX increased (Extended Data Fig. 15a, b). Meanwhile, after ACY738 treatment, we observed a significant decrease in the levels of ISG genes in cells reconstituted with cGAS T68E/S213D compared to that reconstituted with cGAS WT (Extended Data Fig. 15c). These results indicate that ACY738 acts as an inducer of ARIH1, further confirming that ARIH1-mediated activation of the STING pathway occurs primarily through the ARIH1-DNA-PKcs-cGAS axis, rather than through the accumulation of DNA damage.

The phospho-mimetic mutant T68E/S213D of cGAS inhibits the activation of STING pathway mediated by ACY738.

Reviewer #3 (Remarks to the Author): with expertise in cancer immunology/immunotherapy

Summary:

Breast cancer is typically insensitive to ICB therapy. The authors find that cisplatin treatment can sensitize 4T1 breast cancer to ICB therapy and implicate ARIH1 induction. KD of ARIH1 induction is sufficient to sensitize tumors to ICB in a STING-dependent manner. Drug screen identified ACY738 as a potent inducer of ARIH1 that can sensitize 4T1 to ICB and at the doses utilized in mice, is better tolerated than cisplatin.

Overall, with caveats noted below, the authors have done a complete job in investigating ARIH1 a tumor-intrinsic STING-dependent mediator of ICB efficacy. This points to a potentially clinically actionable target to enhance tumor immunogenicity. Whether this applies to other tumor models and the overlap with other reported mechanisms of immunogenic cell death will be the subject of future investigations.

Response:

We thank the reviewer for their appreciation of our study and the constructive comments. Following the suggestions, we have conducted additional experiments and added a total of 50 new experimental panels to the manuscript. The newly added content is highlighted in red text. We hope that our response sufficiently addresses

the concerns raised by the reviewer. Please note that the figure citations in our response below refer to the new (post-revision) figures.

Our study sheds light on the mechanism of overcoming resistance to checkpoint blockade that is regulated by the upregulation of ARIH1 and its associated T cell-mediated effects. ARIH1 can be a promising target for improving the clinical response to ICB therapy, thus providing a new strategy for cancer treatment.

As per reviewer's suggestions, we have now added two *in vivo* experiments further supporting our model.

Whether this applies to other tumor models?

We thank the reviewer for this comment. We have validated our phenotype in additional tumor models, supporting ARIH1 as a promising target for broad-spectrum cancer therapy. We have now performed ARIH1 overexpression in tumor models on immunocompetent mice, including E0771 and B16-F10, and added the data to Extended Data Figure 6a-j and below.

In vitro proliferation of E0771 and B16-F10 cells was not affected by overexpression of the ARIH1 gene (Extended Data Fig. 5a). In the E0771 and B16-F10 models, tumor growth of ARIH1^{WT-OE} tumor-bearing mice treated with anti-PD-L1 was also significantly decreased compared to the wild-type group treated with IgG or anti-PD-L1 (Extended Data Fig. 6a-c, Extended Data Fig. 6f-h), accompanied by increased numbers of infiltrating CD8⁺ T cells and GzmB⁺CD8⁺ T cells (Extended Data Fig. 6d, e, Extended Data Fig. 6i, j). However, treating ARIH1^{C355S-OE} tumors with anti-PD-L1 showed no reduction in tumor growth or no accumulation of tumor-infiltrating CD8⁺ and GzmB⁺CD8⁺ T cells (Extended Data Fig. 6a-j). These results indicate that ARIH1 enhances PD-L1 blockade-induced anti-tumor immunity for a broad spectrum of cancers.

Overexpression of ARIH1 shows no effect on the proliferation of tumor cells *in vitro*.

Extended Data Figure 5a: Tumor cells were infected with an empty vector (CTRL) or

Arih1 overexpressing lentiviral preparations (Arih1^{WT}-OE). Cell viability was monitored at indicated time points by an ATP assay. n=5/group. Data **a** is presented as mean ± SEM, Two-way ANOVA test. ns, not significant.

The E3 ligase activity is required for ARIH1 enhancing PD-L1 blockade-induced anti-tumor immunity.

Extended Data Figure 6a-e: **a.** Tumor growth curves of CTRL, Arih1^{WT}-OE, and Arih1^{C355S}-OE E0771 cells (8×10⁵) in C57BL/6 mice (n=7-8 per group) with indicated treatments. **b-c.** Representative image of tumors (**b**) and tumor weights (**c**) of the mice as in (**a**) at Day20 with the indicated treatments. n=7-8 mice/group. **d-e.** Quantification of tumor-infiltrating CD8⁺ T cells (**d**) and GzmB⁺CD8⁺ T cells (**e**) of the mice as in (**a**). n=7, 7, 5, 8 mice/group. For **a** data is presented as mean ± SEM, Two-way ANOVA test. For **c-e** data are presented as means ± SEM, One-way ANOVA test. **P < 0.01, ****P < 0.0001, ns, not significant.

Extended Data Figure 6f-j: **f.** Tumor growth curves of CTRL, Arih1^{-WT}-OE, and Arih1^{-C355S}-OE B16-F10 cells (5×10^5) in C57BL/6 mice (n=6-10 per group) with indicated treatments. **g-h.** Representative image of tumors (**g**) and tumor weights (**h**) of the mice as in (**f**) at Day17 with the indicated treatments. n=6-10 mice/group. **i-j.** Quantification of tumor-infiltrating CD8⁺ T cells (**i**) and GzmB⁺CD8⁺ T cells (**j**) of the mice as in (**f**). n=6, 6, 3, 10 mice/group. For **f** data is presented as mean \pm SEM, Two-way ANOVA test. For **i-j** data are presented as means \pm SEM, One-way ANOVA test. For **h** data is presented as mean \pm SEM, unpaired Student's t-test. *P < 0.05, **P < 0.01, ***P < 0.001, ****P < 0.0001, ns, not significant.

Major points:

Figure 1-2: The data are compelling and clear. However, for the key in vivo expts (A, H), there is no mention of how many times an expt was performed. In general, it is best practice to repeat in vivo tumor growth experiments with 3 total experiments.

Response:

We thank the reviewer for this comment. We showed three independent experiments of Figure 1-2 (a, h) and have added the description in Figure legends as below:

Data shown in a and h are representative of three independent experiments.

Cisplatin enhances the efficacy of anti-PD-L1 antibody.

Fig 1a

Figure 1a: Tumor growth curves upon subcutaneous injection of 5×10^5 4T1 cells into BALB/c mice treated with vehicle, cisplatin alone, anti-PD-L1 alone and cisplatin+anti-PD-L1. Data shown above represent three independent experiments, n=5-6 mice/group. For **a** data is presented as mean \pm SEM, Two-way ANOVA test. *P < 0.05, **P < 0.01, ****P < 0.0001.

ARIH1 knockdown reverses the anti-tumor effect of PD-L1 plus Cisplatin.

Figure 1h: Tumor growth of Ctrl-KD and Arih1-KD 4T1 cells in BALB/c mice with indicated treatments. Data shown above represent three independent experiments, n=7 mice/group. For h data is presented as mean \pm SEM, Two-way ANOVA test. **P < 0.01, ****P < 0.0001.

ARIH1 enhances the efficacy of anti-PD-L1 antibody.

Figure 2a. Tumor growth curves in BALB/c mice with control (CTRL) and Arih1-overexpressing (Arih1^{WT-OE}) tumors treated with PD-L1 or isotype mAbs intraperitoneally (i.p.) starting on day 7 and then every three days after subcutaneous inoculation of 5×10^5 4T1 cells. Data shown above represent three independent experiments, n=6-7 mice/group. For a data is presented as mean \pm SEM, Two-way ANOVA test. ****P < 0.0001.

ARIH1 enhances the efficacy of anti-PD-L1 antibody dependent on its enzymatic activity.

Figure 2h. Representative tumor growth in BALB/c mice bearing CTRL, Arih1^{WT-OE},

and Aih1^{-C355S}-OE 4T1 tumors with the indicated treatments. Data shown above represent three independent experiments, n=5-7 mice/group. For **h** data is presented as mean \pm SEM, Two-way ANOVA test. ****P < 0.0001, ns, not significant.

Figure 6: Does ACY738 work in the Sting KD tumors?

Response:

We thank the reviewer for this comment. We have now performed STING knockdown in the 4T1 tumor model with ACY738-PD-L1 inhibitor combination and added the data to Extended Data Figure 14c-g and below.

We knocked down Sting in 4T1 cells. The growth of the Sting knockdown tumors was not altered upon combination treatment of ACY738 and PD-L1 blockade (Extended Data Fig. 14c-e). Consistently, Sting knockdown in tumors significantly abrogated the combination treatment-induced CD8⁺ T and GzmB⁺CD8⁺ cell infiltration (Extended Data Fig. 14f-g). These results indicate that the antitumor effect of the combination of ACY738 and PD-L1 blockade is dependent on STING.

Knockdown of STING reverses the anti-tumor effect of PD-L1 blockade with ACY738.

Extended Data Figure 14c-g: **c-e.** Tumor growth curves (**c**), final tumor image (**d**) and tumor weights (**e**) in Ctrl-KD and Sting-KD 4T1 cells in BALB/c mice (n=8 per group) with indicated treatments. **f-g.** Quantification of tumor-infiltrating CD8⁺ T cells (**f**) and GzmB⁺CD8⁺ T cells (**g**) in Ctrl-KD and Sting-KD 4T1 tumors of the mice as in (**c**). n=8 mice/group. For **c** data is presented as mean \pm SEM, Two-way ANOVA test. For **f-g** data are presented as means \pm SEM, One-way ANOVA test. For **e** data is

presented as mean \pm SEM, unpaired Student's t-test. *P < 0.05, **P < 0.01, ***P < 0.001, ****P < 0.0001, ns, not significant.

Minor points:

Line 88: Authors write on line 88 that they used “4T1-derived murine TNBC xenografts”. This is likely a typo since 4T1 into BALB/c is syngeneic. Xenografts is also written throughout the figure legends. If the data are derived from true xenografts, then all of these experiments would be problematic since immunotherapy treatment in humans cannot be modeled in settings of xenogeneic mismatch.

Response:

We thank the reviewer for this comment. We have now amended the text and figure legends and methods of mouse tumor models.

Line 89: No corrections requested, but for what it is worth, other groups have shown sensitivity of 4T1 to PD-1 blockade. See Zappasodi, Merghoub et al. Nat Medicine 2019.

Response:

We thank the reviewer for this comment. We have carefully read the paper by Zappasodi and Merghoub and gained valuable information from it.

Fig 4E: “CRTL” is not spelled correctly.

Response:

We thank the reviewer for this comment. We have now amended the spelling in Fig 4E.

REVIEWERS' COMMENTS

Reviewer #1 (Remarks to the Author):

The authors have done a nice job to address my concerns. I have no more comments and hope to see the impact of their work in the literature.

Reviewer #2 (Remarks to the Author):

The authors have extensively revised the manuscript and addressed all my comments satisfactorily.

Reviewer #3 (Remarks to the Author):

Well done. The manuscript is substantially strengthened with the additions.

Reviewer #1 (Remarks to the Author):

The authors have done a nice job to address my concerns. I have no more comments and hope to see the impact of their work in the literature.

Response:

We thank the reviewer for their appreciation of our study and the constructive comments.

Reviewer #2 (Remarks to the Author):

The authors have extensively revised the manuscript and addressed all my comments satisfactorily.

Response:

We thank the reviewer for their appreciation of our study and the constructive comments.

Reviewer #3 (Remarks to the Author):

Well done. The manuscript is substantially strengthened with the additions.

Response:

We thank the reviewer for their appreciation of our study and the constructive comments.